# HIERARCHICAL SUBSPACES OF POLICIES FOR CONTINUAL OFFLINE REINFORCEMENT LEARNING

## ABSTRACT

In dynamic domains such as autonomous robotics and video game simulations, agents must continuously adapt to new tasks while retaining previously acquired skills. This ongoing process, known as Continual Reinforcement Learning, presents significant challenges, including the risk of forgetting past knowledge and the need for scalable solutions as the number of tasks increases. To address these issues, we introduce HIerarchical LOW-rank Subspaces of Policies (HILOW), a novel framework designed for continual learning in offline navigation settings. HILOW leverages hierarchical policy subspaces to enable flexible and efficient adaptation to new tasks while preserving existing knowledge. We demonstrate, through a careful experimental study, the effectiveness of our method in both classical MuJoCo maze environments and complex video game-like simulations, showcasing competitive performance and satisfying adaptability according to classical continual learning metrics, in particular regarding memory usage. Our work provides a promising framework for real-world applications where continuous learning from pre-collected data is essential.

## 1 INTRODUCTION

Humans continuously acquire new skills and knowledge, adapting to an ever-changing world while retaining what they have previously learned. Designing systems capable of replicating this lifelong learning ability is a key challenge in the Continual Reinforcement Learning (CRL) (Khetarpal et al., 2022) community. Traditional Reinforcement Learning (RL) (Sutton & Barto, 2018), while powerful, often struggles with adaptive, cumulative learning. In CRL, a learning agent must sequentially solve tasks, requiring to master new skills without degrading the knowledge gained from previous tasks.

Within this framework, we focus on a specific subset of problems that combines goal-conditioned learning and offline training, with a particular emphasis on navigation. Goal-Conditioned RL (GCRL) (Ding et al., 2019; Liu et al., 2022a) involves learning policies that can be conditioned to reach specific goal states, making it especially relevant for real-world applications in robotics and video games where navigation is crucial. The *offline* setting (Levine et al., 2020; Prudencio et al., 2023)], which relies on pre-collected datasets is particularly appealing when data collection is expensive, risky, or impractical. However, alone, this setting is not sufficient in the context of changing environments: agents need to continuously adapt to new tasks without forgetting the previous ones, while maintaining scalability as the number of tasks increases (Graffieti et al., 2022; Shaheen et al., 2022).

Various CRL methods have been proposed to tackle these challenges : some use replay buffer or generative models to replicate past tasks (Rolnick et al., 2019; Huang et al., 2021) ; others involve architectural revisions to mitigate forgetting (Rusu et al., 2016; Veniat et al., 2020) ; and some use regularization techniques to improve scalability (Kirkpatrick et al., 2017; Kumar et al., 2023). Nevertheless, these approaches face limitations : Replay-based methods can be impractical due to data storage constraints and privacy concerns, particularly in industries like video game development, where data retention may be costly. Regularization techniques struggle with highly diverse changes, and architecture modifications, such as expanding neural network structures, can become memory-intensive thus limiting scalability. While entirely addressing all these limitations is challenging, Continual Subspace of Policies (CSP) (Gaya et al., 2023) stands out within this literature as an interesting balance between flexibility and efficiency. CSP introduces subspaces of neural networks (Wortsman et al., 2021; Gaya et al., 2022), allowing new parameters to be added when necessary,

which helps adapting without forgetting previous skills. However, CSP is primarily an online method, leveraging Soft Actor-Critic (Haarnoja et al., 2018) as its backbone algorithm, and remains untested in offline settings where it may face new challenges.

In this article, we propose **HIerarchical LOW-Rank Subspaces of Policies (HILOW)**, a practical offline adaptation of Continual Subspace of Policies (CSP) for hierarchical architectures, which is particularly well suited for navigation tasks. HILOW relies on growing separate parameter subspaces, for a high-level path-planner policy and a low-level path-follower policy, depending on the task stream (see *Figure 1*). To properly assess the relevance of HILOW and catalyze further research, we present a comprehensive study of existing methods, introducing new environments and tasks that address the lack of established benchmarks for Continual Offline Reinforcement Learning regarding Goal-Conditioned navigation tasks. While *Section 2* reviews the relevant and related literature, *Section 3* present the theoretical background that contextualizes our research. In *Section 4*, we detail our proposed approach. *Sections 5.1* and *5.2* presents our experimental methodology, comparing our approach in both novel video-game-like settings with human-authored datasets and classical goal-conditioned environments. Finally, *Sections 5.3* to *5.5* present experimental results, evaluating performance across diverse task sequences with standard CRL metrics.

Our main contributions are :

- HILOW, a novel hierarchical framework for Continual Offline RL, leveraging low-rank subspaces of policies for scalable low-memory adaptation for Goal-Conditioned navigation tasks.

- A large panel of Goal-Conditioned navigation tasks associated with datasets, encompassing both robotics and video game scenarios with human-authored datasets. We hope this new open-source benchmark will provide a comprehensive testing ground for future research in this domain.

- A comprehensive experimental evaluation of HILOW and state-of-the-art CRL methods using our proposed benchmark. Our results demonstrate competitive scalability and adaptability of HILOW, showcasing its ability to handle diverse and complex task sequences across various classical metrics.

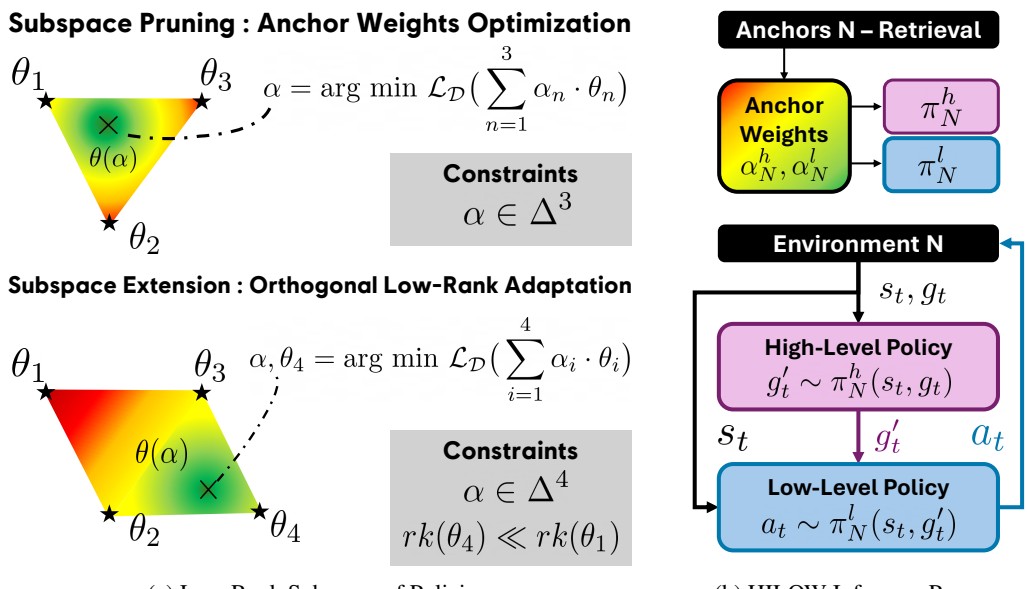

(a) Low-Rank Subspace of Policies.     (b) HILOW Inference Process.

Figure 1: **HIerarchical LOW-Rank subspaces of policies (HILOW)**. (a) Illustration of the pruning and extension mechanisms. *Pruning* involves optimizing anchor weights $\alpha$ within a defined simplex, allowing efficient exploration of the existing subspace. *Extending* introduces new low-rank anchors to expand the subspace, facilitating the adaptation to new tasks while keeping a compact representation. (b) The inference pipeline for task N-*th* leverages learned anchor weights. The high-level policy generates sub-goals, which the low-level policy uses to produce specific actions.

## 2    RELATED WORK

The following section reviews related frameworks and methods to distinguish between various approaches. While these frameworks share common ground, their settings differ in essential ways. By comparing them, we position our work to better highlight the unique challenges we address.

**Transfer Learning** (Da Silva & Costa, 2019; Zhu et al., 2023), along with **Multitask Learning** (Zhang & Yang, 2018; Vithayathil Varghese & Mahmoud, 2020), and **Meta-Learning** (Yu et al., 2020; Gupta et al., 2018; Beck et al., 2023) are foundational paradigms in machine learning that aim to leverage knowledge from multiple tasks to improve learning efficiency and performance. While these approaches excel at leveraging previously acquired knowledge to enhance learning on new tasks, they may require simultaneous access to all tasks during training and do not inherently address the sequential nature of CRL (Table 1). Thus, their use in continual learning scenarios is limited.

| Learning Framework | Tasks Availability | Data Access | Adaptation Paradigm |
|---|---|---|---|
| *Transfer Learning* | Sequential | Real-time interactions or Pre-collected datasets | Learning new tasks, allowed to forget |
| *Multitask Learning* | Simultaneous | Real-time interactions or Pre-collected datasets | Learning multiple tasks together |
| *Meta Learning* | Sequential | Real-time interactions or Pre-collected datasets | Learning how to learn new tasks |
| *Online CRL* | Sequential | Real-time interactions | Learning new tasks without forgetting |
| *Offline CRL* | Sequential | Pre-collected datasets | Learning new tasks without forgetting |

Table 1: Comparison of learning frameworks.

**Continual Reinforcement Learning (CRL)** aims to develop agents capable of learning tasks without forgetting previously acquired knowledge (Khetarpal et al., 2022; Díaz-Rodríguez et al., 2018). CRL methods address the challenge of sequential task learning through various strategies. **Replay-Based** methods mitigate forgetting by storing past experiences and replaying them (Rolnick et al., 2019; Huang et al., 2021). While effective, they can be impractical due to significant storage requirements and potential privacy concerns, especially in industrial applications. **Regularization** techniques like Elastic Weight Consolidation (EWC) (Kirkpatrick et al., 2017) and L2 regularization (Kumar et al., 2023) introduce constraints on parameter updates, but they may struggle with highly diverse tasks. **Architectural** approaches such as Progressive Neural Networks (PNNs) (Rusu et al., 2016) modify a networks architecture to accommodate new tasks. Although these methods isolate task-specific parameters, they can become memory-intensive and lack scalability as the number of tasks grows.

Most CRL research focuses on the **Online Setting**, where agents learn by interacting with the environment (Wang et al., 2024). In contrast, **Offline CRL** involves learning solely from fixed datasets (Isele & Cosgun, 2018; Liu et al., 2024). While Offline CRL is more practical when data collection is demanding, it presents unique challenges due to the inability to gather new data. Benchmark environments are crucial for evaluating and comparing CRL methods. Existing benchmarks like *Continual World* (Wolczyk et al., 2021) or *CORA* (Powers et al., 2022) focus on the online setting and do not specifically address goal-conditioned tasks. To our knowledge, there are no standardized offline benchmarks for **Goal-Conditioned** CRL (Liu et al., 2022b; Park et al., 2024).

Among architectural approaches certain methods, such as **Continual Subspace of Policies (CSP)** (Gaya et al., 2022; 2023) or **Low-Rank Adaptation (LoRA)** (Hu et al., 2021), are relevant for both online and offline settings. CSP uses policies based on a subspace of neural networks (Wortsman et al., 2021). However, it has been primarily applied in online settings, leveraging algorithms like Soft Actor-Critic (Haarnoja et al., 2018), and has not been extensively explored in an offline context. LoRA enhances neural network adaptability by introducing low-rank updates to weight matrices, significantly reducing the number of parameters required for each new task. LoRA has been applied in transfer learning (Hu et al., 2021), multitask learning (Liu et al., 2023), and continual learning for supervised tasks (Wistuba et al., 2023). Its potential for memory efficiency makes it suitable for both online and offline CRL applications, but its integration into offline CRL remains limited.

**Hierarchical Policies (HP)** structure the decision-making process into multiple levels. In CRL, HP facilitate learning by allowing separate components to focus on different aspects of a task. However, existing approaches often rely on complex models, such as large language models (Pan et al., 2024),

or are tailored to meta-learning and multitask learning tasks (Shu et al., 2018; Chua et al., 2023). Our approach integrates HP into an offline CRL framework, enabling lightweight flexible adaptation.

Despite the aforementioned advancements, our work stands out by specifically addressing the unique challenges of Offline CRL, without relying on data retention strategies. To the best of our knowledge, no benchmarks exist for Offline CRL in Goal-Conditioned navigation settings, underscoring the novelty of our framework and the importance of our introduced benchmark.

## 3 PRELIMINARIES

In this section, we present the necessary background to understand our approach and the problem it solves. This includes formal definitions and key concepts related to Markov Decision Processes, Goal-Conditioned Reinforcement Learning, and Continual Reinforcement Learning.

We consider a Markov Decision Process (MDP) $\mathcal{M} = \big( \mathcal{S}, \mathcal{A}, \mathcal{P}_{\mathcal{S}}, \mathcal{P}_{\mathcal{S}}^{(0)}, \mathcal{R}, \gamma \big)$, which provides a formal framework for RL, where $\mathcal{S}$ is a state space, $\mathcal{A}$ an action space, $\mathcal{P}_{\mathcal{S}} : \mathcal{S} \times \mathcal{A} \to \Delta(\mathcal{S})$ a transition function, $\mathcal{P}_{\mathcal{S}}^{(0)} \in \Delta(S)$ an initial distribution over the states, $\mathcal{R} : \mathcal{S} \times \mathcal{A} \times \mathcal{S} \to \mathbb{R}$ a deterministic reward function, and $\gamma \in ]0, 1]$ a discount factor. An agent's behavior follows a policy $\pi_\theta : \mathcal{S} \to \Delta(\mathcal{A})$, parameterized by $\theta \in \Theta$. The objective is to learn optimal parameters $\theta_{\mathcal{M}}^*$ maximizing the expected cumulative reward $J_{\mathcal{M}}(\theta)$ or the success rate $\sigma_{\mathcal{M}}(\theta)$.

**Offline Goal-Conditioned RL**   We extend the MDP to include a goal space $\mathcal{G}$, introducing $\mathcal{P}_{\mathcal{S},\mathcal{G}}^{(0)}$ an initial state and goal distribution, $\phi : \mathcal{S} \to \mathcal{G}$ a function mapping each state to the goal it represents, and $d : \mathcal{G} \times \mathcal{G} \to \mathbb{R}^+$ a distance metric on $\mathcal{G}$. The policy $\pi_\theta : \mathcal{S} \times \mathcal{G} \to \Delta(\mathcal{A})$ and the reward function $\mathcal{R} : \mathcal{S} \times \mathcal{A} \times \mathcal{S} \times \mathcal{G} \to \mathbb{R}$ are now conditioned on a goal $g \in \mathcal{G}$. We consider sparse rewards allocated when the agent reaches the goal within a range $0 \leq \epsilon : \mathcal{R}(s_t, a_t, s_{t+1}, g) = \mathbb{1}\big( d(\phi(s_{t+1}), g) \leq \epsilon \big)$. Given a dataset $\mathcal{D} = \big\{ (s, a, r, s', g) \big\}$, the policy loss is optimized to reach the specified goals. This formulation represents our core problem addressed when facing a new task.

**Continual Reinforcement Learning**   In CRL, an agent follows a sequence of tasks, or stream, $\mathcal{T} = \big( T_1, ..., T_N \big)$, with $T_k = \mathcal{M}_k$ or $T_k = (\mathcal{M}_k, \mathcal{D}_k)$. We note $\theta_k$ the parameters of a policy after learning on the $k$-th task. As the agent learns new skills, it must either preserve (to prevent forgetting) or enhance (to encourage backward transfer) its performance on tasks already learned, while ideally having a relatively low number of parameters. To quantitatively compare CRL methods, we adopt standard metrics commonly used in the literature (Díaz-Rodríguez et al., 2018; Kemker et al., 2018) :

**Performance (PER) :** $\frac{1}{N} \sum_{k=1}^{N} \sigma_{\mathcal{M}_k}(\theta_N)$; **Backward Transfer (BWT) :** $\frac{1}{N} \sum_{k=1}^{N} \big( \sigma_{\mathcal{M}_k}(\theta_N) - \sigma_{\mathcal{M}_k}(\theta_k) \big)$;

**Forward Transfer (FWT) :** $\frac{1}{N} \sum_{k=1}^{N} \big( \sigma_{\mathcal{M}_k}(\theta_k) - \sigma_{\mathcal{M}_k}(\tilde{\theta}_k) \big)$ ; **Relative Model Size (MEM) :** $\frac{|\theta_N|}{|\theta_{\text{ref}}|}$.

The performance metric measures the average success rate across all tasks. Backward transfer indicates how learning a new task affects previous ones, while forward transfer measures the ability to transfer knowledge to new tasks, using $\tilde{\theta}_k$ as randomly initialized parameters. The relative model size compares the memory load of the model to a reference model associated to parameters $\theta_{\text{ref}}$ .

**Subspace of Neural Networks**   A subspace of neural networks is a convex hull within the space of parameters (Wortsman et al., 2021). Building one involves finding a finite set of anchors that serve as a basis. In formal terms, given a high-dimensional parameter space $\Theta$, a subspace $\mathcal{V}(\theta_1, \ldots, \theta_n) \subset \Theta$ is defined by a set of anchor points $\{\theta_1, \theta_2, \ldots, \theta_k\} \subset \Theta$. These points form the basis of the subspace, and any point $\theta \in \mathcal{V}(\theta_1, \ldots, \theta_n)$ can be represented as a linear combination of these anchors :

$$\theta = \sum_{i=1}^{n} \alpha_i \theta_i \quad \text{where} \quad \alpha \in \Delta^k, \quad \text{i.e.} \quad \sum_{i=1}^{n} \alpha_i = 1 \quad \text{and} \quad \alpha_i \in \mathbb{R}^+ \tag{1}$$

Exploring the subspace involves adjusting anchor weights $\alpha_i$ within a lower-dimensional space. If needed, new anchors extend the subspace, expanding its capacity while preserving prior knowledge. Unlike previous methods, we use two subspaces : a high-level and a low-level one, offering flexibility by avoiding unnecessary expansions (e.g. not extending the high-level subspace if only low-level adjustments are needed). Additionally, we propose a new policy evaluation procedure – which conditions subspace expansion – to better fit offline learning settings (see *Section 4.3* for details).

**Low-Rank Adaptation (LoRA)** extends neural networks for new tasks by approximating updates to weight matrices using low-rank structures. Given a trained weight $W \in \mathbb{R}^{n \times m}$, we adapt it by introducing an update $W' = W + \Delta W$, where $\Delta W$ is a low-rank approximation. Specifically, $\Delta W$ is factored as $A \in \mathbb{R}^{n \times r}$ and $B \in \mathbb{R}^{r \times m}$, with $r \ll \min(m, n)$. In our framework, LoRA is used to generate new anchors for the subspaces, where they can be expressed as $\theta_i = A_i B_i$, when $i \geq 2$.

## 4 HIERARCHICAL SUBSPACE OF POLICIES

We now provide a detailed description of HIerarchical LOW-Rank Subspaces of Policies (HILOW). *Section 4.1* introduces the Hierarchical Imitation Learning algorithm, the backbone of our approach. Next, *Section 4.2* provides a high-level overview of the core learning steps involved in HILOW. We then cover low-rank subspace extension in *Section 4.3*, and subspace exploration in *Section 4.4*. See *Algorithm 1* for a detailed pseudo-code about learning a subspace of policies in an offline setting.

### 4.1 HIERARCHICAL IMITATION LEARNING

Hierarchical Imitation Learning (Gupta et al., 2019) forms the backbone of our approach for each given task, by learning both high-level and low-level policies using a provided dataset of episodes $\mathcal{D} = \left\{ (s_t^i, a_t^i, r_t^i, s_{t+1}^i, g^i) \right\}$. The overall policy is parameterized by $\theta = (\theta_h, \theta_l)$, where $\theta_h$ governs the high-level policy and $\theta_l$ controls the low-level one. This structure allows the agent to break down complex tasks into simpler ones, facilitating both long-term planning and short-term action execution.

- **High-Level Policy Training :** The high-level policy is trained to predict a sub-goal $\phi(s_{t+k})$, where $k$ is the *waystep* hyperparameter determining how far into the future the sub-goal is :

$$\mathcal{L}_{\mathcal{D}}^h(\theta_h) = \mathbb{E}_{(s_t^i, s_{t+k}^i, g^i) \sim \mathcal{D}} \left[ -\log(\pi_{\theta_h}^h(\phi(s_{t+k}^i)|s_t^i, g^i)) \right]$$

- **Low-Level Policy Training :** The low-level policy $\pi^l$ is trained to execute actions that take the agent towards the sub-goals proposed by the high-level policy :

$$\mathcal{L}_{\mathcal{D}}^l(\theta_l) = \mathbb{E}_{(s_t^i, a_t^i, s_{t+1}^i, \phi(s_{t+k}^i)) \sim \mathcal{D}} \left[ -\log(\pi_{\theta_l}^l(a_t|s_t^i, \phi(s_{t+k}^i))) \right]$$

- **Hindsight Experience Replay (HER) (Andrychowicz et al., 2017; Packer et al., 2021) :** We perform data augmentation using HER, which relabels the goal of a given transition with the goal representation of a future state within the same trajectories considered.

### 4.2 HILOW LEARNING ALGORITHM : OVERVIEW

The HILOW Learning Algorithm manages hierarchical policies through distinct subspaces, each specializing to different aspects of task adaptation. This specialization promotes both efficiency and scalability, allowing our framework to handle diverse and sequential tasks in an offline setting.

**Initial Anchor Training :** We begin by training the initial anchor parameters $\theta_1^h \in \Theta^h$ and $\theta_1^l \in \Theta^l$ on the first task $T_1$. The anchor weights $\alpha_1^h \in \Delta^1$ and $\alpha_1^l \in \Delta^1$ are set to $(1)$, indicating reliance on the initial anchors. This establishes the foundational subspaces for high-level and low-level policies.

**Training on Subsequent Tasks :** For each new task $T_k$ and for each of the two considered subspaces, the algorithm performs the following steps to efficiently adapt the learning policy :

1. **Subspace Extension :** We introduce low-rank parameters $\theta_{N^h+1}^h \in \Theta_r^h$ and $\theta_{N^l+1}^l \in \Theta_r^l$, where $r$ is the rank, and initialize new anchor weights $\alpha_{\text{curr}}^h$ and $\alpha_{\text{curr}}^l$. These new anchors and anchors weights are then learned from the dataset $\mathcal{D}_k$ using Hierarchical Imitation Learning.

2. **Previous Subspace Exploration and Evaluation :** We explore different anchor weights by sampling from a Dirichlet distribution with equal weights, to uniformly search over the previous subspace. Each sampled configuration is evaluated on a few batches from the new task's dataset $\mathcal{D}_k$, and the one minimizing the loss is selected as a representative of the previous subspace.

3. **Subspace Adaptation Decision:** We compare the loss of the extended subspace ($L_{\text{curr}}$) with that of the previous subspace ($L_{\text{prev}}$) given a criterion $\epsilon > 0$. Considering positive losses, if $L_{\text{prev}} \leq (1 \pm \epsilon) \cdot L_{\text{curr}}$, we prune the new anchor, retaining the previous subspace configuration. Otherwise, we retain the new anchor, effectively accommodating the subspace to the new task.

---

**Algorithm 1 Offline Learning of a low-Rank Subspace of Policies**

---

**Require:** stream $\mathcal{T}$ ; number of epochs E ; learning rate $\eta$ ; criterion $\epsilon$ ; rank $r$ ; sample size $S$ .

1:  **Train initial anchors :**
2:  Initialize anchor parameters $\theta_1 \sim \Theta$ and anchor weights $\alpha_1 \leftarrow (1)$
3:  **for** $epoch = 1$ to E **do**
4:      Batched gradient descent : $\theta_1 \leftarrow \theta_1 - \eta \nabla \mathcal{L}_{\mathcal{B}}(\alpha_{1,1} \cdot \theta_1)$

5:  **Train subsequent anchors :**
6:  **for** $k = 2$ to $\texttt{len}(\mathcal{T})$ **do**
7:      Consider $N$ previously trained high anchor parameters $\theta_1, \dots, \theta_N$
8:      **Train $k$-th anchor :**
9:      Initialize anchor parameters $\theta_{N+1} \sim \Theta_r$ and anchor scores $\hat{\alpha}_{\text{curr}} \leftarrow (0, \dots, 0) = 0_{N+1}$
10:     **for** $epoch = 1$ to E **do**
11:         **for** mini-batch $\mathcal{B}$ in $\mathcal{D}_k$ **do**
12:             Compute anchor weight : $\alpha_{\text{curr}} \leftarrow \texttt{softmax}(\hat{\alpha}_{\text{curr}})$
13:             Update $\theta_{N+1}$ using gradient descent : $\theta_{N+1} \leftarrow \theta_{N+1} - \eta \nabla \mathcal{L}_{\mathcal{B}}(\sum_{i=1}^{N+1} \alpha_{\text{curr},i} \cdot \theta_i)$
14:             Update $\hat{\alpha}_{\text{curr}}$ using gradient descent : $\hat{\alpha}_{\text{curr}} \leftarrow \hat{\alpha}_{\text{curr}} - \eta \nabla \mathcal{L}_{\mathcal{B}}(\sum_{i=1}^{N+1} \alpha_{\text{curr},i} \cdot \theta_i)$
15:     **Evaluate current subspace** (*Section 4.3*) :
16:     Compute current anchor weight : $\alpha_{\text{curr}} \leftarrow \texttt{softmax}(\hat{\alpha}_{\text{curr}})$
17:     Compute current loss : $L_{\text{curr}} \leftarrow \mathcal{L}_{\mathcal{D}_k} \left( \sum_{i=1}^{N+1} \alpha_{\text{curr},i} \cdot \theta_i \right)$
18:     **Find optimal weights for previous subspace** (*Section 4.4*) :
19:     Sample $S$ anchor weights $\{\alpha'^{(s)}\}_{s=1}^{S} \sim \text{Dirichlet}(\mathbf{1}_N)$
20:     Set $\alpha_{\text{prev}} \leftarrow \arg\min_{\alpha'^{(s)}} \mathcal{L}_{\mathcal{D}_k} \left( \sum_{i=1}^{N} \alpha'_i \cdot \theta_i \right)$
21:     Compute $L_{\text{prev}} \leftarrow \mathcal{L}_{\mathcal{D}_k} \left( \sum_{i=1}^{N} \alpha_{\text{prev},i} \cdot \theta_i \right)$
22:     **Criterion based adaptation decision :**
23:     **if** $L_{\text{prev}} \leq (1 \pm \epsilon) \cdot L_{\text{curr}}$ **then**
24:         **Pruning :** $\alpha_k \leftarrow \alpha_{\text{prev}}$, discard $\theta_{N+1}$
25:     **else**
26:         **Extending :** $\alpha_k \leftarrow \texttt{softmax}(\hat{\alpha}_{\text{curr}})$, keep $\theta_{N+1}$

---

## 4.3 EXTENDING A SUBSPACE

Extending a subspace involves integrating a new anchor parameter $\theta_{N+1}$ and its corresponding weight $\alpha_{\text{curr}}$ into the existing set of anchors. This process allows the model to incorporate task-specific variations while retaining the ability to leverage previously learned policies.

Initially, $\theta_{N+1}$ is randomly initialized, and anchor scores $\hat{\alpha}_{\text{curr}} = (0, \dots, 0)$ are set to zeros. During training, the softmax function is applied to the anchor scores, yielding the anchor weights $\alpha_{\text{curr}}$. This step ensures that the weights are positive and sum to one, providing smooth and differentiable control over how much each anchor contributes to the final policy.

The learning process proceeds by updating both $\theta_{N+1}$ and $\hat{\alpha}_{\text{curr}}$ using gradient descent [1]. Specifically, $\theta_{N+1}$ and $\hat{\alpha}_{\text{curr}}$ are updated by minimizing the learning loss over the dataset using mini-batches $\mathcal{B}$, by considering the weighted contributions $\alpha_{\text{curr}}$ :

$$\theta_{N+1} \leftarrow \theta_{N+1} - \eta \nabla \mathcal{L}_{\mathcal{B}} \left( \sum_{i=1}^{N+1} \alpha_{\text{curr},i} \cdot \theta_i \right) \ , \ \hat{\alpha}_{\text{curr}} \leftarrow \hat{\alpha}_{\text{curr}} - \eta \nabla \mathcal{L}_{\mathcal{B}} \left( \sum_{i=1}^{N+1} \alpha_{\text{curr},i} \cdot \theta_i \right)$$

In practice, whenever a new anchor is added, the anchor weights for previous tasks are extended by appending a zero to the weight vector. This ensures that the dimensionality of weight vectors is consistent across all tasks : $\alpha_i \leftarrow (\alpha_i, 0), \quad \forall i \in \{1, \dots, N\}$ .

This approach allows the model to efficiently reuse knowledge from previously learned tasks while adjusting to the specific requirements of the new one. By adding the new anchor, the subspace is expanded, enabling the model to handle a broader range of tasks without forgetting previous skills.

---

[1]In contrast to CSP (Gaya et al., 2023) which relies on sampling anchor weights for both policy pruning and extension. Nevertheless, we do sample them when evaluating the previous subspace for better flexibility.

## 4.4 EXPLORING A SUBSPACE

After training the new anchor, we evaluate whether the subspace should be extended or pruned. This decision is based on a comparison between the loss of the current extended subspace (including the new anchor) and the loss of the previous subspace (without the new anchor).

To compute the current subspace loss, we use the learned $\alpha_{\mathrm{curr}}$ : $L_{\mathrm{curr}} = \mathcal{L}_{\mathcal{D}_k}\left(\sum_{i=1}^{N+1} \alpha_{\mathrm{curr},i} \cdot \theta_i\right)$. This loss measures how well the newly extended subspace performs on the task's dataset. For the previous subspace, we aim to find the weights $\alpha_{\mathrm{prev}}$ that minimize the loss over the previous anchors $\theta_1, \ldots, \theta_N$, excluding the newly added anchor $\theta_{N+1}$. In theory, this would involve finding $\alpha_{\mathrm{prev}} = \arg\min_\alpha \mathcal{L}_{\mathcal{D}_k}\left(\sum_{i=1}^{N} \alpha_i \cdot \theta_i\right)$. However, in practice, performing a full optimization over $\alpha$ can be computationally expensive. Instead, we sample $S$ weight vectors $\alpha'$ from a Dirichlet distribution over the simplex $\Delta^N$ and compute the corresponding loss for each sample:

$$\alpha' \sim \mathrm{Dirichlet}(\Delta^N) \quad, \quad L'_{\mathrm{prev}} = \mathcal{L}_{\mathcal{D}_k}\left(\sum_{i=1}^{N} \alpha'_i \cdot \theta_i\right) \quad \text{and} \quad L_{\mathrm{prev}} = \min_{\alpha'} L'_{\mathrm{prev}}$$

This approach provides a computationally efficient approximation to the full optimization problem by leveraging random sampling from the Dirichlet distribution.

Once both losses are computed, the decision to prune or extend the subspace is made based on a predefined criterion. If the previous subspace loss $L_{\mathrm{prev}}$ is within an acceptable range of the current subspace loss $L_{\mathrm{curr}}$, the new anchor $\theta_{N+1}$ is pruned, and the anchor weights are reverted to the best previous configuration $\alpha_{\mathrm{prev}}$. Specifically : $L_{\mathrm{prev}} \leq (1 \pm \epsilon) \cdot L_{\mathrm{curr}}$ . On the other hand, if the extended subspace performs significantly better, the subspace is retained, and the weights $\alpha_{\mathrm{curr}}$ are kept.

## 5 EXPERIMENTS

Our experiments aim to address the following questions : How does HILOW compare to relevant baselines in terms of performance and memory metrics (Section 5.3) ? How does it perform in terms of forgetting and generalization metrics (Section 5.4) ? Lastly, we explore through an ablation study (Section 5.5) : How do the core design principles of HILOW affect its performance ?

## 5.1 ENVIRONMENTS & TASK STREAMS

We consider multiple scenarios designed to test the ability to adapt and transfer knowledge between tasks. These experiments span two types of environments : classical maze benchmarks from the Gymnasium framework and custom video game-like environments implemented in Godot (see *Figure 2*). Details about these environments and the considered streams are provided in the *Appendix A*.

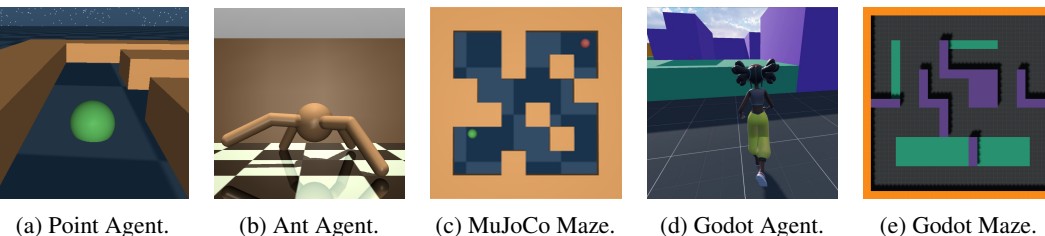

| (a) Point Agent. | (b) Ant Agent. | (c) MuJoCo Maze. | (d) Godot Agent. | (e) Godot Maze. |

Figure 2: **Point Agent** is a point mass controlled by applying forces in two dimensions. **Ant Agent** is a more complex 8-DoF articulated quadruped robot controlled by torques. **Godot Agent** is a 3D character controlled by both continuous and discrete actions replicating video game controls.

The classical maze environments (Lazcano et al., 2023), PointMaze and AntMaze, are well-known in deep learning but less explored in the CRL. We introduce a novel use of those by customizing datasets and environments from Minari (Younis et al., 2024) to create task variations such as inverse actions or permuted observations. We also introduce more complex maze-like 3D navigation environments in Godot, *SimpleTown* and *AmazeVille*, which feature topological changes across task streams with human-authored datasets, reflecting the evolving nature of game worlds in the video game industry.

We assess the performance of the different approaches on a diverse set of task streams, with randomly generated sequences, which also tests the agent's capacity to adapt across unpredictable transitions.

## 5.2 Continual Reinforcement Learning Baselines

We compare our method to several CRL strategies relevant to our setting, as described in *Section 3*. All baselines are built on the same Hierarchical Imitation Learning backbone and detailed in *Appendix A.4*.

The **Single Naive Strategy (SC1)** trains a single policy from scratch on the latest dataset and applies it to all tasks, while the **Expanding Naive Strategy (SCN)** trains and saves a new policy for each task. The **Single Finetuning Strategy (FT1)** adapts a single policy across tasks but suffers from catastrophic forgetting. In contrast, the **Expanding Finetuning Strategy (FTN)** retains a separate policy for each task, preserving knowledge but increasing memory use. The **Freeze Strategy (FZ)** trains a policy on the first task and applies it unchanged to all subsequent tasks. More advanced methods include **L2-Regularization (L2)** (Kumar et al., 2023), which adds a penalty to the loss function according to the previous weight changes between tasks, and **Elastic Weight Consolidation (EWC)** (Kirkpatrick et al., 2017), which supposedly improves L2 by penalizing important weights using the Fisher Information Matrix. **Progressive Neural Networks (PNN)** (Rusu et al., 2016) add new layers for each task, using lateral connections to transfer useful representations while avoiding interference. Finally, we adapt **Continual Subspace of Policies (CSP)** (Gaya et al., 2023), originally designed for online learning, for offline use while maintaining Q-function learning.

## 5.3 Performance and Relative Memory Size

The trade-off between performance and memory usage is critical in CRL. *Figure 3* illustrates the average Performance (PER) according to the Relative Memory Size (MEM) of the baseline strategies and ours. HILOW consistently demonstrates high performance with moderate memory consumption, outperforming or matching other methods in this balance.

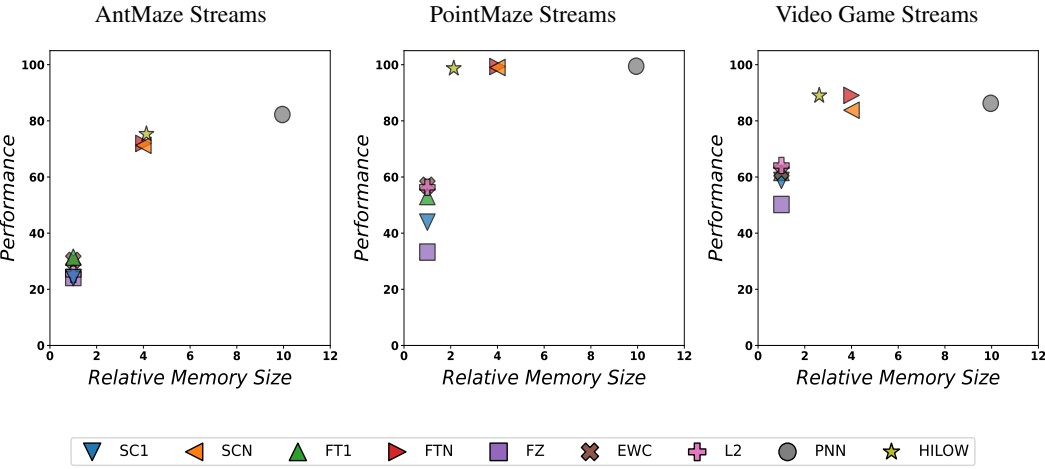

Figure 3: **Performance vs. Relative Memory Size.** The figure shows the average performance w.r.t. memory size of different CRL methods over sets of streams from our three considered environments. **HILOW** (yellow star) demonstrates **high performance** with **moderate memory usage**.

In the **AntMaze streams**, our **HILOW** method approaches the top-performing one **PNN** while using significantly less memory. The simple architectural strategies like **FTN** and **SCN** perform slightly below **HILOW** with comparable memory consumption. In contrast, weight regularization and naive methods (e.g., **EWC**, **FT1**, **FZ**) underperform in both metrics. These results demonstrate that **HILOW** effectively balances performance and resource use[2]. In the **PointMaze streams**, **HILOW** nearly matches the top-performing **PNN**, while maintaining significantly lower memory usage.

---

[2]For complex tasks like AntMaze, starting with a slightly larger model enhances HILOW's low-rank adaptors, resulting in a marginally larger final model than FTN and SCN.

Simple architectural methods (**FTN** and **SCN**) show high task performance but require more memory compared to **HILOW**. The weight regularization and naive strategies, as in AntMaze, fail to provide comparable performance, which highlights the advantage of **HILOW** in memory-constrained environments. In the **Video Game streams**, **HILOW** surpasses **PNN** both in performance and relative memory size. It remains highly competitive with **FTN**, which matches **HILOW**'s performance but at the cost of more memory.

Overall, the **HILOW** method consistently demonstrates its strong performance across diverse tasks while maintaining a significantly lower memory usage, especially when compared to memory-heavy methods like **PNN**, which has an exponential memory cost (see figure 4). This balance makes **HILOW** a highly efficient approach for continual reinforcement learning in resource-constrained environments.

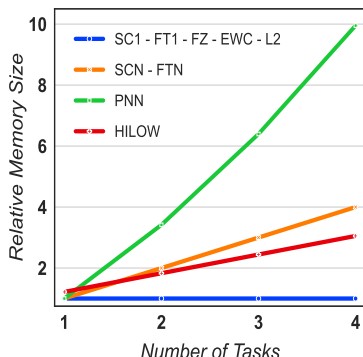

Figure 4: Evolution of the Relative Memory Size metric according to the number of tasks.

### 5.4 FORGETTING AND GENERALIZATION

Table 2: **Performance Related Metrics.** Backward Transfer (BWT) and Forward Transfer (FWT) across methods and streams. Architectural approaches like **FTN**, **SCN**, **PNN** and **HILOW** excel in BWT by preventing forgetting through parameter storage.

| Method | AntMaze Streams | | | PointMaze Streams | | | Video Game Streams | | |
|---|---|---|---|---|---|---|---|---|---|
| | PER ↑ | BWT ↑ | FWT ↑ | PER ↑ | BWT ↑ | FWT ↑ | PER ↑ | BWT ↑ | FWT ↑ |
| SC1 | 24.2 | -49.0 | 0.0 | 43.9 | -55.1 | 0.0 | 58.8 | -25.6 | 0.0 |
| SCN | 71.3 | **0.0** | 0.0 | **99.0** | **0.0** | 0.0 | 83.8 | **0.0** | 0.0 |
| FT1 | 31.4 | -47.2 | 5.5 | 53.0 | -46.3 | **0.3** | 61.6 | -27.6 | 4.9 |
| FTN | 72.0 | **0.0** | 5.5 | **99.3** | **0.0** | **0.3** | 89.1 | **0.0** | 4.9 |
| FZ | 24.2 | **0.0** | -52.1 | 33.3 | **0.0** | -65.7 | 50.2 | **0.0** | -33.1 |
| L2 | 25.3 | -43.8 | -4.0 | 56.3 | -41.2 | -1.4 | 64.1 | -18.2 | -2.1 |
| EWC | 30.3 | -47.0 | 4.2 | 57.1 | -42.0 | 0.1 | 61.9 | -28.1 | **5.6** |
| PNN | **82.3** | **0.0** | **9.1** | **99.5** | **0.0** | 0.5 | **86.2** | **0.0** | 1.7 |
| HILOW | 75.3 | **0.0** | 2.2 | **98.7** | **0.0** | -0.8 | **89.0** | **0.0** | 4.6 |

**Table 2** summarizes the Backward Transfer (BWT) and Forward Transfer (FWT) metrics for the different methods across AntMaze, PointMaze, and Video Game streams. In general, architectural methods like **FTN**, **SCN**, **PNN** and **HILOW** perform well in terms of **BWT**, as they can store task-specific parameters without overwriting previous ones, allowing them to avoid forgetting. On the other hand, weight regularization methods (**EWC**, **L2**) can struggle when task changes are more diverse, showing inconsistent BWT results. Regarding **forward transfer (FWT)**, most methods exhibit minimal or no forward transfer, highlighting the inherent challenge of knowledge transfer between tasks. While methods such as **FT1** and **FTN** demonstrate some positive forward transfer, **HILOW** shows only modest improvements. This may indicate limitations in the low-rank adaptor's capacity for task generalization, particularly in dynamic environments. Although **HILOW** does not excel in forward transfer metrics, it maintains a stable and balanced performance across tasks, making it a robust option for effectively managing memory and performance in continual learning settings.

### 5.5 ABLATIONS

To understand the effectiveness of our proposed HILOW framework, we conduct a series of ablation studies on PointMaze streams, featuring methods ranging from vanilla adaptations of CSP to HILOW (see Table 3).

**Adapting CSP to Offline CRL : CSP-O**   The original Continual Subspace of Policies (CSP) (Gaya et al., 2023) leverages Soft Actor-Critic (SAC) and replay buffers to evaluate policies by learning a Q-function. However, in an offline setting, replay buffers are impractical. To adapt CSP for offline

CRL, we replace SAC with Hierarchical Imitation Learning (HBC) and eliminate the reliance on replay buffers, creating a single subspace that encapsulates both high-level and low-level policy parameters. We refer to this approach as *CSP* in Table 3. Despite these modifications, we observe that the Q-function's predictions become nearly independent of actions due to the optimality of expert data, limiting CSP's effectiveness in offline scenarios. To address the limitations of the adapted CSP, we develop CSP-O, an improved offline adaptation that employs a loss function-based selection criterion instead of a Q-function. CSP-O maintains a single subspace but enhances policy evaluation by directly minimizing the loss with respect to the expert behavior, thereby improving performance in offline settings.

**Why Two Subspaces ?** While CSP-O consolidates all policy parameters into a single subspace, our **HILOW** divides them into two distinct subspaces: one for high-level and one for low-level policies. This separation enables more fine-tuned updates, preventing unnecessary expansions of the high-level subspace when only low-level adjustments are needed. Our experiments show that HILOW outperforms CSP-O in PointMaze streams.

| CSP Method | PointMaze Streams | |
|---|---|---|
| | PER ↑ | MEM ↓ |
| CSP | $64.3 \pm 98.2$ | $2.5 \pm 0.5$ |
| CSP-O | $99.1 \pm 0.6$ | $4.0 \pm 0.0$ |
| HILOW (w/o LoRA) | $98.8 \pm 0.3$ | $3.5 \pm 0.5$ |
| HILOW | $98.7 \pm 0.1$ | $2.1 \pm 0.2$ |

Table 3: HILOW ablations on Point-Maze Streams.

**Benefits of Low-Rank Adaptation** Low-Rank Adaptation (LoRA) enables efficient parameter updates by approximating changes with low-rank matrices. Comparing HILOW with and without LoRa subspaces, we find that incorporating low-rank adaptors allows for smaller, more efficient updates when adapting to new tasks.

# 6 DISCUSSION

In this work, we introduced **HILOW**, a framework that combines hierarchical imitation learning with low-rank subspace adaptations for offline continual reinforcement learning. Our results show that our framework effectively balances performance and memory usage across diverse environments, including classical mazes and complex video games. By using separate subspaces for high-level and low-level policies, it efficiently adapts to new tasks while mitigating forgetting. Compared to other methods, **HILOW** offers a strong trade-off between adaptability and resource efficiency.

Future work could study to which extent HILOW can scale to more complex CRL settings, e.g. with chaotic task streams. Additionally, while HILOW performs well with expert data, scenarios with imperfect expert trajectories could pose challenges. Towards such settings, improved subspace evaluations procedured could be studies, e.g. integrating *Inverse Reinforcement Learning* (IRL) (Arora & Doshi, 2021; Ho & Ermon, 2016), to refine Q-function learning, providing better scoring of sampled policies. Exploring adaptive ranks during subspace extension could enhance the framework's flexibility and scalability with task complexity. Overall, **HILOW** makes a significant step toward addressing offline continual learning challenges.

# 7 REPRODUCIBILITY STATEMENT

We have made extensive efforts to ensure that our work is fully reproducible. Detailed descriptions of the environments, tasks streams, and training settings are provided in the *Section 5*, and the *Appendix A* to *C*, including specifics for the different environments. We also include pseudo-code for all algorithms, covering both our proposed HILOW framework and the baselines, with a thorough explanation of Hierarchical Imitation Learning, the backbone of our approach. To further facilitate replication, we will share the full source code with reviewers and release it publicly, alongside the data, upon acceptance. This code include all necessary components for running experiments, training models, and evaluating metrics. Additionally, all theoretical assumptions, parameters settings, and decision processes, such as subspace pruning and regularization, are clearly documented to ensure clarity and replicability.

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

## A    TASK STREAMS DETAILS

### A.1    ENVIRONMENTS

#### A.1.1    MUJOCO MAZE ENVIRONMENTS

We consider two sets of environments from the Gymnasium framework Lazcano et al. (2023) : PointMaze and AntMaze. They are considered due to their complexity and the availability of datasets from D4RL Fu et al. (2020), which provide a standardized set of tasks to evaluate CRL algorithms.

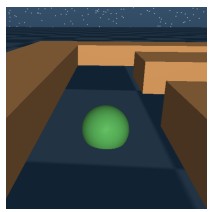
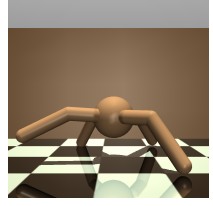
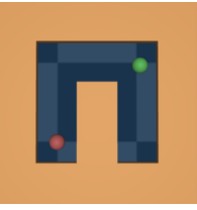
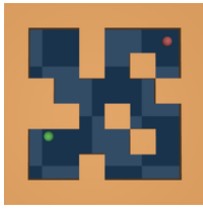
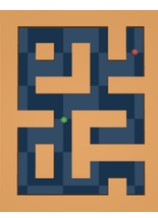

(a) Point Agent.        (b) Ant Agent.        (c) U Maze.        (d) Medium Maze.    (e) Large Maze.

Figure 5: All `U` (`size` $= 5 \times 5$), `M` (`size` $= 8 \times 8$), and `L` (`size` $= 12 \times 9$) mazes provide a sparse reward with a value of $1$ when the agent is within a $0.5$ unit radius to the goal. The **Point Agent** is a point mass controlled by applying forces in two dimensions, allowing the agent to move freely across the plane towards a goal location. In contrast the **Ant Agent** is a more complex articulated quadruped robot. It is controlled through the application of torques to its joints.

#### A.1.2    VIDEO GAME NAVIGATION ENVIRONMENTS

While PointMaze and AntMaze environments were simple to setup and allowed us to quickly generate datasets, as to our knowledge there are no CRL datasets for navigation, they are primarily focused on assessing the impact of changes in agent dynamics, such as action transformations. These environments are expressive but lack features needed to fully understand how topographic variations affect an agent. To bridge this gap, We introduce a video-game like 3D navigation environments, implemented on Godot (Godot (2020)), that offer diverse mazes with more explainable spatial challenges. They allow us to explore the influence of environmental structures on agent performance.

There are two families of mazes : **SimpleTown**, which mazes are relatively simple, with a size of $30 \times 30$ meters. The starting positions are randomly sampled on one side, and the goal positions are on the other side ; **AmazeVille**, which mazes are more challenging, with a size of $60 \times 60$ meters. They have a finite set of start and goal positions, and include two subsets of maps : some with high blocks, *i.e.* not jumpable obstacles ; others with low blocks, *i.e.* jumpable ones.

| Observation Feature | Size | Type | Observation Feature | Size | Type |
|---|---|---|---|---|---|
| Agent Position | 3 | `float` | Goal Position | 3 | `float` |
| Agent Orientation | 3 | `float` | Agent Velocity | 3 | `float` |
| RGB Image | $3 \times 64 \times 64$ | `float` | Depth Image | $11 \times 11$ | `float` |
| Floor Contact | 1 | `bool` | Wall Contact | 1 | `bool` |
| Goal Contact | 1 | `bool` | Timestep | 1 | `int` |
| Up Direction | 3 | `float` | - | - | - |

Table 4: **(Godot) Available observation features.** The maximum number of features an observation may have is $12440$, if it were to use all the available ones. The position information correspond to the $(x, y, z)$ coordinates in meters. The agent orientation is its angle in radian according to the vertical axis. The velocity is provided in meters per second. The RGB images corresponds to the visualization of the environment from the agent's point field of view. The depth image is obtained using $11 \times 11$ raycasts from the agent position to the visible nearest obstacles.

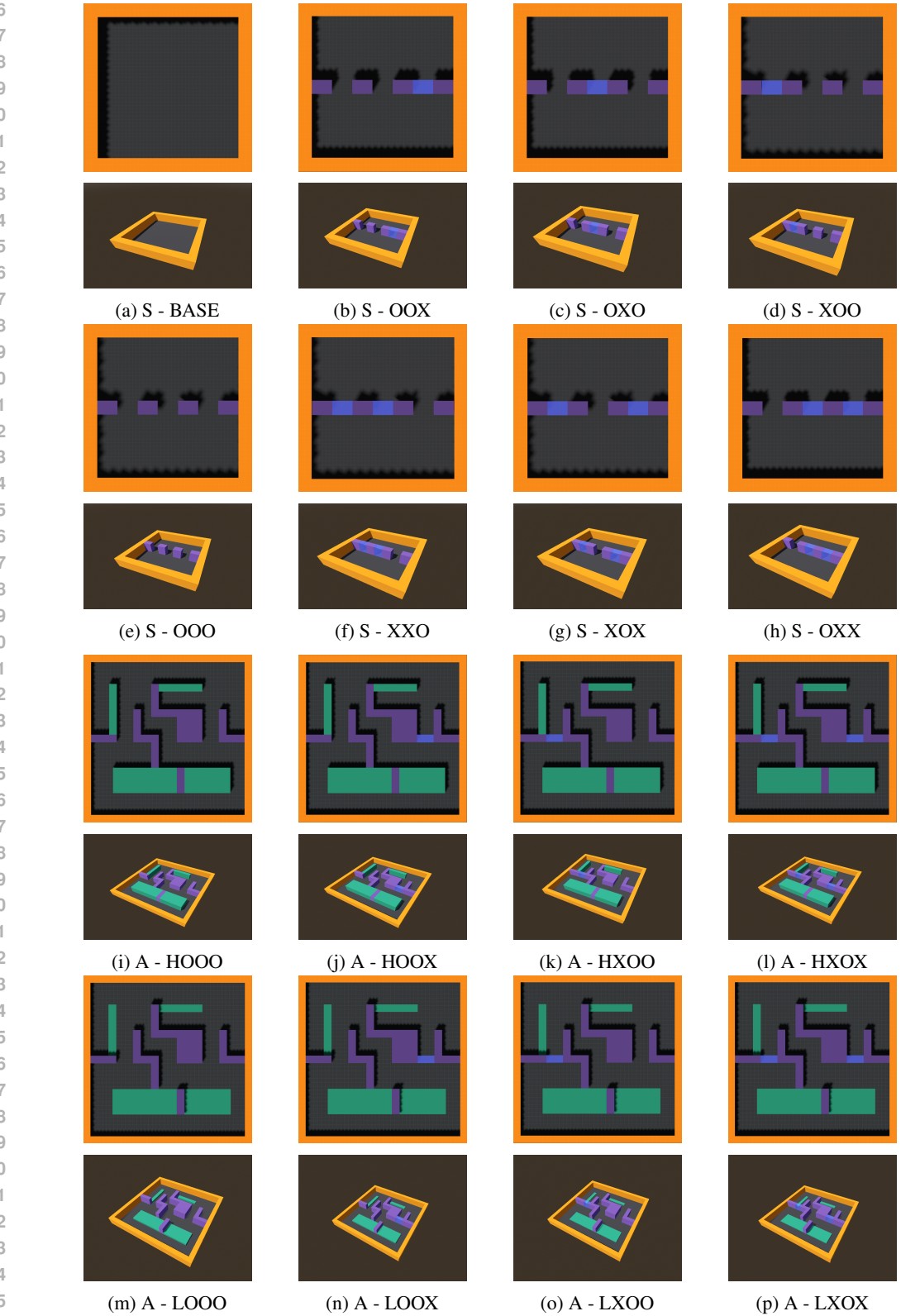

Figure 6: **The SimpleTown (S) and the AmazeVille (AH, AL) environments :** The naming indicate whether specific doors are open (O) or not (X), and if movable green blocks are in high positions (H) or low positions (L), providing a clear way to distinguish between different maze configurations.

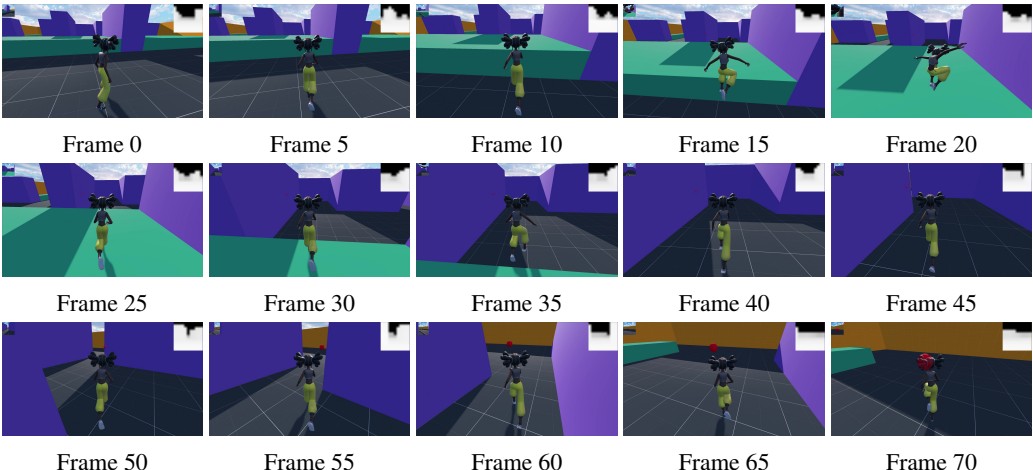

Figure 7: **Visualization of a Human-Generated Trajectory on A - LOOO.**

## A.2 TASKS

We design a variety of tasks within each environment to evaluate the agent's adaptability to different scenarios. For both PointMaze and AntMaze environments, we consider five task variations :

- Normal (N) : The standard task with no changes to actions or observations.
- Inverse Actions (IA) : Opposing values of the action features.
- Inverse Observations (IO) : Opposing values of the observation features.
- Permute Actions (PA) : clockwise permutation of the actions features.
- Permute Observations (PO) : Clockwise permutation of the observation features.

For the Godot-based environments (SimpleTown and AmazeVille), we simply use the mazes provided without additional modifications. The inherent complexity of these mazes, including variations in obstacle placement, already presents a significant challenge for the learning algorithms.

## A.3 DATASETS

For the PointMaze and AntMaze environments, we employed datasets from D4RL, each comprising 500 episodes per task across different maze configurations. Due to the straightforward nature of the task transformations, we effectively adapted the original datasets by applying these modifications and developed corresponding environment wrappers for seamless integration within the Gym framework.

The trajectories visualized in Figures 8 and 9 illustrate not only the richness and diversity of the collected data but also the complexity of the tasks that agents must navigate. These trajectories highlight a range of behaviors, from straightforward goal-reaching paths to more intricate maneuvers required to overcome environmental obstacles.

In the Godot-based environments, data was sampled manually over approximately 10 hours, resulting in 100 episodes for each AmazeVille maze and 250 episodes for each SimpleTown maze.

## A.4 TASK STREAMS

A task stream refers to a sequence of environments and corresponding datasets that an agent learns from over time. Each task in the stream introduces new environmental variations, changes in dynamics, or modifications to the observation and action spaces, simulating the possible evolving challenges in real-world scenarios. They may build upon previously learned skills, testing both short-term adaptability and long-term memory retention. We consider several classical metrics, namely : Performance (PER), Backward Transfer (BWT), Forward Transfer (FWT), and Relative Memory Size (MEM). These metrics enable us to assess the agent's continual learning capabilities by evaluating its ability to generalize across tasks, preserve learned knowledge, while being scalable.

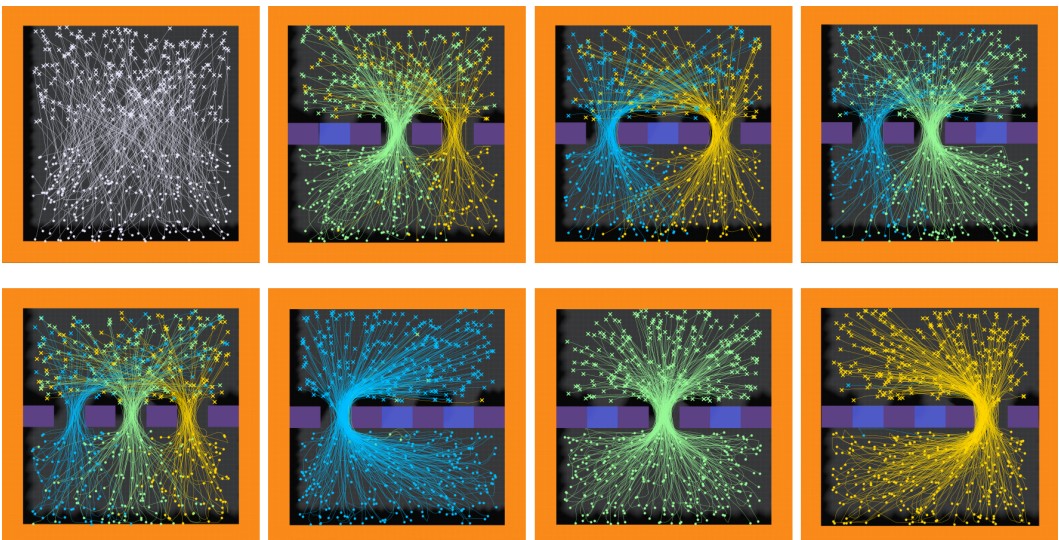

Figure 8: SimpleTown Trajectories (Staring Position are on the bottom).

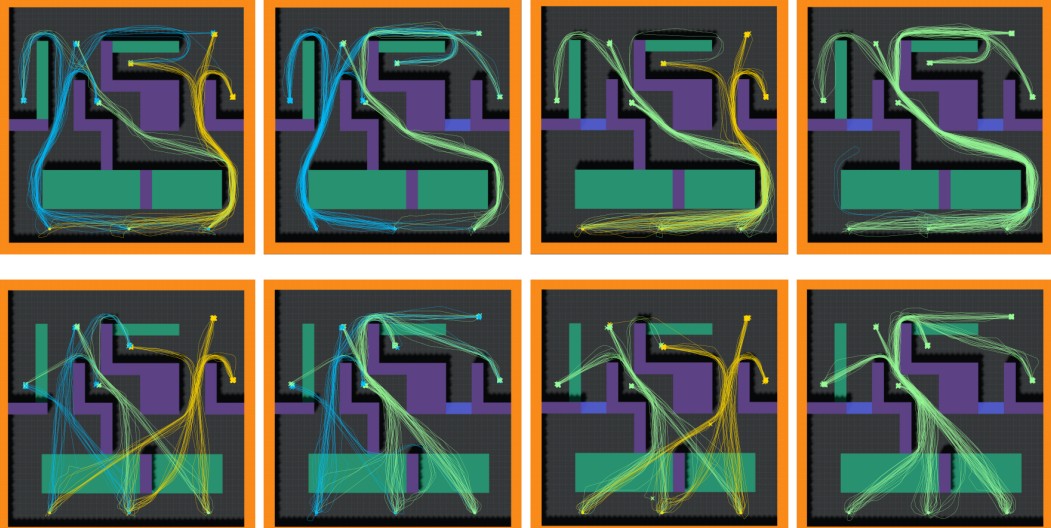

Figure 9: AmazeVille Trajectories (Staring Position are on the bottom).

Here are the AntMaze streams (`maze-task` $[n_{\text{episodes}}]$) :

- `1 : U-N[500]` $\rightarrow$ `L-N[500]` $\rightarrow$ `U-PO[500]` $\rightarrow$ `M-IO[500]`
- `2 : U-PA[500]` $\rightarrow$ `M-PO[500]` $\rightarrow$ `M-N[500]` $\rightarrow$ `M-N[500]`

Here are the PointMaze streams (`maze-task` $[n_{\text{episodes}}]$) :

- `1 : U-N[500]` $\rightarrow$ `L-N[500]` $\rightarrow$ `U-PO[500]` $\rightarrow$ `M-IO[500]`
- `2 : U-PA[500]` $\rightarrow$ `M-PO[500]` $\rightarrow$ `M-N[500]` $\rightarrow$ `M-N[500]`

Here are the Video Game streams (`maze` $[n_{\text{episodes}}]$) :

- `1 : HOOO[100]` $\rightarrow$ `HXOO[100]` $\rightarrow$ `LOOO[100]` $\rightarrow$ `LOOX[100]`
- `2 : LOOX[100]` $\rightarrow$ `HXOO[100]` $\rightarrow$ `HOOO[100]` $\rightarrow$ `LXOX[100]`

## B    BASELINES DETAILS

### B.1    GOAL-CONDITIONED OFFLINE REINFORCEMENT LEARNING ALGORITHMS

In both Imitation Learning and Hierarchical Imitation Learning algorithms, we will consider a MDP $\mathcal{M} = \big( \mathcal{S}, \mathcal{A}, \mathcal{P}_{\mathcal{S}}, \mathcal{P}_{\mathcal{S},\mathcal{G}}^{(0)}, \mathcal{R}, \gamma, \mathcal{G}, \phi, d \big)$, and a dataset of pre-collected trajectories $\mathcal{D} = \big\{ (s_t^i, a_t^i, r_t^i, s_{t+1}^i, g^i) \big\}$ sampled by one or many expert agents.

**Imitation Learning (BC) Ding et al. (2019).**    The BC algorithm is a simple framework to leverage a dataset of transitions $\mathcal{D}$ by running a supervised regression using a negative log-likelihood loss :

$$\mathcal{L}_{\mathcal{D}}(\theta) = \mathbb{E}_{(s_t^i, a, s_t^i, r, s_t^i, s_{t+1}^i, g^i) \sim \mathcal{D}} \Big[ - log(\pi_\theta(a_t^i | s_t^i, g^i)) \Big] \ , \ \text{and} \ \ \theta_{\mathcal{D}}^* = \underset{\theta \, \in \, \Theta}{\arg \min} \, \mathcal{L}_{\mathcal{D}}(\theta) \qquad (2)$$

Moreover this algorithm benefit from using a HER (Figure 10) relabelling strategy. Indeed, as the trajectories have been sampled by an expert, if we consider a transition $(s_t^i, a_t^i, r_t^i, s_{t+1}^i, g^i) \in \mathcal{D}$ then we can also consider $(s_t^i, a_t^i, r_t^i, s_{t+1}^i, \phi(s_{t+k}^i))$ as also an expert generated transition. Thus, HER can be considered as a data augmentation technique, which is particularly effective in low data regime.

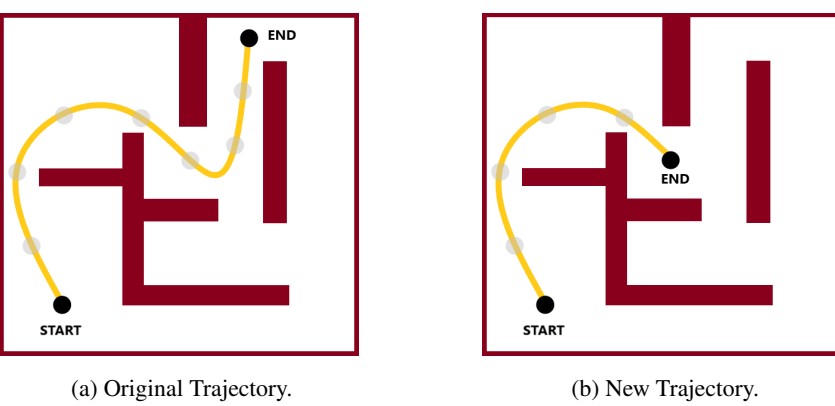

(a) Original Trajectory.                                (b) New Trajectory.

Figure 10: Hindsight Experience Replay (HER) Illustration.

**Hierarchical Imitation Learning (HBC) (Le et al., 2018; Gupta et al., 2019; Park et al., 2023).**
HBC leverages hierarchical structures so as to effectively handle the challenges associated with learning from offline datasets. This algorithm decomposes the navigation task into manageable sub-tasks using a high-level and a low-level policy.

Now, an end-to-end policy $\pi_\theta : \mathcal{S} \times \mathcal{G} \to \Delta(\mathcal{A})$ is divided into two distinct learnable components. First, a high policy $\pi_{\theta_h}^h : \mathcal{S} \times \mathcal{G} \to \Delta(\mathcal{G})$ aiming at selecting intermediate sub-goals that are strategically feasible stepping stones towards a final goal, thus simplifying the path finding task. Then, a low policy $\pi_{\theta_l}^l : \mathcal{S} \times \mathcal{G} \to \Delta(\mathcal{A})$ focused on generating the actions necessary to progress from the current state towards the sub-goal selected by the high policy. The optimization follows :

$$\mathcal{L}_{\mathcal{D}}^h(\theta_h) = \mathbb{E}_{(s_t^i, s_{t+k}^i, g^i) \sim \mathcal{D}} \Big[ - log(\pi_{\theta_h}^h(\phi(s_{t+k}^i) | s_t^i, g^i))) \Big] \ , \ \text{and} \ \ \theta_h{}_{\mathcal{D}}^* = \underset{\theta_h \, \in \, \Theta}{\arg \min} \, \mathcal{L}_{\mathcal{D}}^h(\theta_h) \qquad (3)$$

$$\mathcal{L}_{\mathcal{D}}^l(\theta_l) = \mathbb{E}_{(s_t^i, a_t^i, s_{t+1}^i, s_{t+k}^i, g^i) \sim \mathcal{D}} \Big[ - log(\pi_{\theta_l}^l(a_t | s_t^i, \phi(s_{t+k}^i))) \Big] \ , \ \text{and} \ \ \theta_l{}_{\mathcal{D}}^* = \underset{\theta_l \, \in \, \Theta}{\arg \min} \, \mathcal{L}_{\mathcal{D}}^l(\theta_l) \qquad (4)$$

Hence, given *way step* hyperparameter $k$, which determines the desired temporal distance of the sub-goals, the optimization for the high and low policies uses a common loss structure, adapted to suit their specific roles.

## B.2 CONTINUAL REINFORCEMENT LEARNING BASELINES

This section explores CRL baselines, designed to learn from a task stream $\mathcal{T}$, where each task $T_k$ consists of a MDP $\mathcal{M}_k = \left( \mathcal{S}_k, \mathcal{A}_k, \mathcal{P}_{\mathcal{S}_k}, \mathcal{P}_{\mathcal{S}, \mathcal{G}_k}^{(0)}, \mathcal{R}_k, \gamma_k, \mathcal{G}, \phi_k, d_k \right)$ and a dataset of trajectories $\mathcal{D}_k = \left\{ (s_t^{k,i}, a_t^{k,i}, r_t^{k,i}, s_{t+1}^{k,i}, g^{k,i}) \right\}$. Interestingly, these strategies could be extended to a broader range algorithms, beyond goal-conditioned ones.

**Naive Learning Strategy or *From Scratch* (`SC1` & `SCN`).** In `SC1`, a single policy is learned from the latest dataset and then applied unchanged to all tasks. In `SCN`, a new policy is trained for each task, improving performance at the cost of a memory load.

---
**Algorithm 2** Naive Strategy

---
**Require:** learning rate $\eta$, number of epochs E, boolean `StorePolicies`
1: **for** $k = 1$ to $N$ **do**
2:     Initialize policy parameters $\theta_k$
3:     **for** $epoch = 1$ to E **do**
4:         **for** mini-batch $\mathcal{B}$ in $\mathcal{D}_k$ **do**
5:             Update $\theta_k$ using gradient descent: $\theta_k \leftarrow \theta_k - \eta \nabla \mathcal{L}_{\mathcal{B}}(\theta_k)$
6:     **if** `StorePolicies` **then** Store $\theta_k$
7:     **else** $\theta_1 \leftarrow \theta_k$

---

**Freeze Strategy (`FZ`).** In the Freeze Strategy, a single policy is trained only on the first task and then applied without modification to all subsequent tasks.

---
**Algorithm 3** Freeze Strategy

---
**Require:** learning rate $\eta$, number of epochs E
1: Initialize policy parameters $\theta_1$
2: **for** $epoch = 1$ to E **do**
3:     **for** mini-batch $\mathcal{B}$ in $\mathcal{D}_1$ **do**
4:         Update $\theta_1$ using gradient descent: $\theta_1 \leftarrow \theta_1 - \eta \nabla \mathcal{L}_{\mathcal{B}}(\theta_1)$

---

**Finetuning Strategy (`FT1` & `FTN`).** The Finetuning Strategy involves adapting a policy learned from the initial task to each subsequent task, either by continuously updating a single policy (`FT1`) or by copying and then updating the policy for each new task (`FTN`), allowing for better task adaptation.

---
**Algorithm 4** Finetuning Strategy

---
**Require:** learning rate $\eta$, number of epochs E, boolean `StorePolicies`
1: Initialize policy parameters $\theta_1$
2: **for** $epoch = 1$ to E **do**
3:     **for** mini-batch $\mathcal{B}$ in $\mathcal{D}_1$ **do**
4:         Update $\theta_1$ using gradient descent: $\theta_1 \leftarrow \theta_1 - \eta \nabla \mathcal{L}_{\mathcal{B}}(\theta_1)$
5: **for** $k = 2$ to $N$ **do**
6:     **if** `StorePolicies` **then** $\theta_k \leftarrow \theta_{k-1}$
7:     **else** $\theta_k \leftarrow \theta_1$
8:     **for** $epoch = 1$ to E **do**
9:         **for** mini-batch $\mathcal{B}$ in $\mathcal{D}_k$ **do**
10:            Update $\theta_k$ using gradient descent: $\theta_k \leftarrow \theta_k - \eta \nabla \mathcal{L}_{\mathcal{B}}(\theta_k)$
11:     **if** `StorePolicies` **then** Store $\theta_k$
12:     **else** $\theta_1 \leftarrow \theta_k$

---

**Elastic Weight Consolidation (`EWC`) Kirkpatrick et al. (2017).** This strategy has been designed to mitigate catastrophic forgetting in continual learning. It achieves this by selectively slowing down learning on certain weights based on their importance to previously learned tasks. This importance is measured by the Fisher Information Matrix, which quantifies the sensitivity of the output function to changes in the parameters.

`EWC` introduces a quadratic penalty to the loss function, constraining the parameters close to their values from previous tasks, where the strength of the penalty is proportional to each parameter's importance. This allows the model to retain performance on previous tasks while continuing to learn new tasks effectively.

However this method struggles for navigation tasks due to the penalty for updating parameters, making it difficult to adapt to tasks like inverse actions. This rigidity is problematic in complex environments where different tasks demand flexibility. As a result, `EWC` is limited in effectively handling tasks requiring greater adaptation.

---

**Algorithm 5** Elastic Weight Consolidation Strategy

---

**Require:** learning rate $\eta$, number of epochs E, elastic weight $\lambda$, Fisher Information Matrix $\mathcal{F}_0$
1: Initialize policy parameters $\theta$
2: **for** $k = 1$ to $N$ **do**
3:      **for** $epoch = 1$ to E **do**
4:          **for** mini-batch $\mathcal{B}$ in $\mathcal{D}_k$ **do**
5:              Compute standard loss : $\mathcal{L}_{\mathcal{B}}^S(\theta)$
6:              Compute EWC loss : $\mathcal{L}_{\mathcal{B}}^{EWC}(\theta) = \frac{\lambda}{2} \sum_{i=1}^{k-1} \mathcal{F}_i \cdot (\theta_i - \theta_{i,\text{old}})^2$
7:              Total loss : $\mathcal{L}_{\mathcal{B}}(\theta) = \mathcal{L}_{\mathcal{B}}^S(\theta) + \mathcal{L}_{\mathcal{B}}^{EWC}(\theta)$
8:              Update $\theta$ using gradient descent: $\theta \leftarrow \theta - \eta \nabla \mathcal{L}_{\mathcal{B}}(\theta)$
9:      Update Fisher Information Matrix $\mathcal{F}_k$
10:     Store current parameters to learn next ones $\theta_{k,\text{old}} \leftarrow \theta$

---

**L2-Regularization Finetuning (L2) Kumar et al. (2023).** This strategy also mitigates catastrophic forgetting by adding an L2 penalty to the loss, discouraging large weight changes during training. This helps preserve knowledge from previous tasks by promoting stability in the learned representations.

As with EWC, L2-regularization struggles in CRL for navigation tasks, especially when actions or dynamics change drastically. The method limits the network's flexibility by forcing small weight updates, making it difficult to adapt to tasks that require distinct actions for similar states, which is critical in evolving environments.

---

**Algorithm 6** L2-Regularization Finetuning Strategy

---

**Require:** learning rate $\eta$, number of epochs E, regularization strength $\lambda$
1: Initialize policy parameters $\theta$ with $\theta$
2: **for** $k = 1$ to $N$ **do**
3:      **for** $epoch = 1$ to E **do**
4:          **for** mini-batch $\mathcal{B}$ in $\mathcal{D}_k$ **do**
5:              Compute task-specific loss : $\mathcal{L}_{\mathcal{B}}^S(\theta)$
6:              Compute L2 regularization loss : $\mathcal{L}_{\mathcal{B}}^{L2}(\theta) = \lambda \|\theta - \theta_{\text{old}}\|^2$
7:              Total loss : $\mathcal{L}_{\mathcal{D}_k}(\theta) = \mathcal{L}_{\mathcal{B}}^S(\theta) + \mathcal{L}_{\mathcal{B}}^{L2}(\theta)$
8:              Update $\theta$ using gradient descent: $\theta \leftarrow \theta - \eta \nabla \mathcal{L}_{\mathcal{B}}(\theta)$

---

**Progressive Neural Networks (PNN) Rusu et al. (2016).** This framework introduce a new column layers for each task, freezing previous weights to preserve knowledge. Lateral connections allow feature transfer, leveraging prior experience while avoiding interference. PNNs effectively prevent catastrophic forgetting, but the model grows with each task, limiting scalability for many tasks or limited memory contexts.

---

**Algorithm 7** Progressive Neural Networks Strategy

---

**Require:** number of tasks $N$, learning rate $\eta$
1: Initialize first task column $C_1$ with random weights
2: Train $C_1$ on the dataset $\mathcal{D}_1$ for the first task
3: **for** $k = 2$ to $N$ **do** ▷ For each new task
4:      Create a new task-specific column $C_k$ with random weights
5:      Freeze weights in previous columns $C_1, C_2, \ldots, C_{k-1}$
6:      Add lateral connections from $C_1, \ldots, C_{k-1}$ to $C_k$
7:      Load task-specific dataset $\mathcal{D}_k$
8:      **for** each mini-batch $\mathcal{B}$ in $\mathcal{D}_k$ **do**
9:          Compute the outputs of previous columns $C_1, \ldots, C_{k-1}$
10:         Pass outputs through lateral connections to $C_k$
11:        Update the weights in $C_k$ using gradient descent
12:      Freeze the weights in column $C_k$ after training

---

**Continual Subspace of Policies (CSP) Gaya et al. (2023).** This strategy handles continual learning by maintaining a subspace of policy parameters that adapt as new tasks are learned. For each new task, a new anchor is added, allowing the model to combine parameters from previous tasks. CSP decides whether to extend or prune the subspace based on a critic, $W_\phi$, that evaluates the performance of anchor combinations.

---

**Algorithm 8** Continual Subspace of Policies (CSP)

---

1: **Input:** $\theta_1, \ldots, \theta_j$ (previous anchors), $\epsilon$ (threshold)
2: **Initialize:** $W_\phi$ (subspace critic), $\mathcal{B}$ (replay buffer)
3: **Initialize:** $\theta_{j+1} \leftarrow \frac{1}{j} \sum_{i=1}^{j} \theta_i$ (new anchor)
4: **for** $i = 1, \ldots, \mathcal{B}$ **do** ▷ // Grow the Subspace
5:      Sample $\alpha \sim \text{Dir}(\mathcal{U}(j+1))$
6:      Set policy parameters $\theta_\alpha \leftarrow \sum_{i=1}^{j+1} \alpha_i \theta_i$
7:      **for** $l = 1, \ldots, K$ **do**
8:          Collect and store $(s, a, r, s', \alpha)$ in $\mathcal{B}$ by sampling $a \sim \pi_{\theta_\alpha}(s)$
9:      **if** *time to update* **then**
10:         Update $\pi_{\theta_{j+1}}$ and $W_\phi$ using the SAC algorithm and the replay buffer $\mathcal{B}$
11:
12: Use $\mathcal{B}$ and $W_\phi$ to estimate: ▷ // Extend or Prune the Subspace
13:

$$\alpha^{\text{old}} \leftarrow \underset{\alpha \in \mathbb{R}_+^n, \|\alpha\|_1 = 1}{\arg\max} W_\phi(\alpha)$$

$$\alpha^{\text{new}} \leftarrow \underset{\alpha \in \mathbb{R}_+^{n+1}, \|\alpha\|_1 = 1}{\arg\max} W_\phi(\alpha)$$

14: **if** $W_\phi(\cdot, \alpha^{\text{new}}) > (1 + \epsilon) \cdot W_\phi(\cdot, \alpha^{\text{old}})$ **then**
15:      **Return:** $\theta_1, \ldots, \theta_j, \theta_{j+1}, \alpha^{\text{new}}$ ▷ // Extend
16: **else**
17:      **Return:** $\theta_1, \ldots, \theta_j, \alpha^{\text{old}}$ ▷ // Prune

---

## C IMPLEMENTATION DETAILS

### C.1 ARCHITECTURES & HYPERPARAMETERS

We primarily followed prior work (Ghosh et al., 2023) for network architectures and hyperparameters. All environments used MLPs with layer normalization on hidden layers. Low-level policies had 256 hidden units, and high-level policies used 64. For HILOW in AntMaze and Godot, we increased these to 300 and 70 respectively as, experimentally, low-rank adaptors performed better with larger initial models on more complex tasks. Dropout of 0.1 was applied to all hidden layers.

Input sizes were 31 for **AntMaze** (including position, goal, and features), 8 for **PointMaze**, and 133 for **Godot**. Output sizes were 8 for AntMaze and Godot, and 2 for PointMaze. Outputs were continuous for AntMaze and PointMaze, while Godot used both continuous and discrete outputs to simulate gamepad controls.

| Hyperparameter | AntMaze | PointMaze | AmazeVille | SimpleTown |
|---|---|---|---|---|
| Batch Size | 1024 | 1024 | 64 | 64 |
| Learning Rate | $3e\text{-}4$ | | | |
| Way Steps (Sub-goal distance) | Umaze : 10 Medium : 15 Large : 15 | Umaze : 50 Medium : 25 Large : 25 | 10 | 3 |
| HER Sampling Temperature | 50.0 | Umaze : 100.0 Medium : 75.0 Large : 100.0 | 100.0 | 15.0 |

Table 5: Hyperparameter settings for AntMaze, PointMaze, and Godot environments.

### C.2 TRAINING DETAILS

For both the EWC and the L2 strategies, we experimented with five different regularization weights $\lambda \in \{ 1e\text{-}2, 1e\text{-}1, 1, 1e1, 1e2 \}$ and selected the *best* model in terms of performance for each task stream. Similarly, for HILOW, we tested different acceptance values $\epsilon \in \{ 1e\text{-}2, 5e\text{-}2, 1e\text{-}1, 2.5e\text{-}1 \}$ to decide whether to prune or extend a subspace.

When using Hierarchical Imitation Learning, we also employed Hindsight Experience Replay (HER) for all environments, using an exponential sampling strategy guided by a temperature parameter to improve sample efficiency.

### C.3 COMPUTE RESOURCES

Training was conducted on a shared compute cluster using CPUs for all experiments, as the models are relatively small and the backbone algorithms do not require highly intensive operations typically associated with GPU use. This choice also allowed us to run more experiments in parallel, optimizing resource utilization. The compute cluster featured Intel(R) Xeon(R) CPU E5-1650 and Intel Cascade Lake 6248 processors. For most models, 4 cores per training were sufficient, but due to PNN's growing memory requirements, we allocated 6 cores for its experiments. Total training times across the defined streams of tasks ranged from 10 to 18 hours, depending on the complexity of the task stream and the run time of the considered CRL strategy.

## D ADDITIONAL & DETAILED RESULTS

### D.1 HIERARCHICAL VS. NON-HIERARCHICAL POLICIES IN GOAL-CONDITIONED RL

*Table 6* compares Imitation Learning and Hierarchical Imitation Learning across the various maze environments. HBC consistently outperforms BC in both success rate and episode length, especially in complex environments like AmazeVille, where hierarchical decision-making is crucial for navigating diverse tasks and obstacles. In simpler environments like SimpleTown, the performance difference is minimal, as these tasks are easier to solve.

| Environment | Maze | Success Rate ↑ | | Episode Length ↓ | |
|---|---|---|---|---|---|
| | | BC | HBC | BC | HBC |
| **PointMaze** | Umaze | $99.2 \pm 1.4$ | $\textbf{100.0} \pm \textbf{0.0}$ | $68.4 \pm 10.9$ | $\textbf{63.8} \pm \textbf{6.2}$ |
| | Medium | $94.1 \pm 8.4$ | $\textbf{99.5} \pm \textbf{1.1}$ | $199.5 \pm 32.2$ | $\textbf{172.0} \pm \textbf{33.1}$ |
| | Large | $67.9 \pm 9.7$ | $\textbf{95.0} \pm \textbf{6.9}$ | $328.5 \pm 33.3$ | $\textbf{282.5} \pm \textbf{61.4}$ |
| **AntMaze** | Umaze | $76.7 \pm 8.5$ | $\textbf{93.5} \pm \textbf{5.4}$ | $422.0 \pm 75.9$ | $\textbf{286.6} \pm \textbf{48.8}$ |
| | Medium | $43.3 \pm 10.5$ | $\textbf{68.8} \pm \textbf{5.0}$ | $688.0 \pm 101.1$ | $\textbf{519.1} \pm \textbf{61.4}$ |
| | Large | $18.8 \pm 11.4$ | $\textbf{32.8} \pm \textbf{9.9}$ | $861.4 \pm 88.9$ | $\textbf{816.8} \pm \textbf{63.8}$ |
| **SimpleTown** | BASE | $94.8 \pm 5.0$ | $\textbf{98.6} \pm \textbf{2.0}$ | $52.7 \pm 3.6$ | $\textbf{51.5} \pm \textbf{2.6}$ |
| | OOO | $95.9 \pm 1.9$ | $\textbf{97.3} \pm \textbf{1.9}$ | $\textbf{55.8} \pm \textbf{2.2}$ | $56.0 \pm 2.5$ |
| | OOX | $92.6 \pm 4.8$ | $\textbf{94.3} \pm \textbf{3.2}$ | $60.6 \pm 2.3$ | $\textbf{59.7} \pm \textbf{4.0}$ |
| | OXO | $89.5 \pm 4.4$ | $\textbf{91.6} \pm \textbf{4.2}$ | $\textbf{61.7} \pm \textbf{1.9}$ | $62.8 \pm 1.2$ |
| | XOO | $\textbf{94.0} \pm \textbf{4.0}$ | $93.8 \pm 3.7$ | $\textbf{59.3} \pm \textbf{3.0}$ | $60.0 \pm 2.4$ |
| | XXO | $\textbf{89.8} \pm \textbf{7.2}$ | $84.2 \pm 5.3$ | $\textbf{70.2} \pm \textbf{2.5}$ | $72.6 \pm 1.7$ |
| | XOX | $90.1 \pm 5.7$ | $\textbf{97.0} \pm \textbf{2.3}$ | $61.4 \pm 2.5$ | $\textbf{60.2} \pm \textbf{1.8}$ |
| | OXX | $\textbf{93.4} \pm \textbf{4.3}$ | $91.3 \pm 3.0$ | $\textbf{67.5} \pm \textbf{0.9}$ | $69.5 \pm 1.6$ |
| **AmazeVille** | HOOO | $70.5 \pm 9.7$ | $\textbf{88.8} \pm \textbf{6.3}$ | $211.0 \pm 12.8$ | $\textbf{182.5} \pm \textbf{9.3}$ |
| | HOOX | $51.2 \pm 13.0$ | $\textbf{78.6} \pm \textbf{8.7}$ | $249.8 \pm 18.9$ | $\textbf{226.0} \pm \textbf{14.2}$ |
| | HXOO | $60.4 \pm 15.8$ | $\textbf{94.8} \pm \textbf{4.7}$ | $228.3 \pm 19.8$ | $\textbf{190.8} \pm \textbf{9.1}$ |
| | HXOX | $46.5 \pm 9.9$ | $\textbf{75.9} \pm \textbf{5.2}$ | $273.7 \pm 11.9$ | $\textbf{240.8} \pm \textbf{4.7}$ |
| | LOOO | $49.6 \pm 3.5$ | $\textbf{75.0} \pm \textbf{7.1}$ | $221.9 \pm 6.0$ | $\textbf{172.2} \pm \textbf{18.0}$ |
| | LOOX | $59.9 \pm 7.2$ | $\textbf{82.9} \pm \textbf{6.3}$ | $225.9 \pm 12.2$ | $\textbf{174.8} \pm \textbf{9.6}$ |
| | LXOO | $47.0 \pm 5.8$ | $\textbf{75.9} \pm \textbf{6.3}$ | $222.8 \pm 8.3$ | $\textbf{169.3} \pm \textbf{13.6}$ |
| | LXOX | $60.1 \pm 8.8$ | $\textbf{95.6} \pm \textbf{4.6}$ | $221.3 \pm 14.8$ | $\textbf{159.9} \pm \textbf{10.1}$ |

Table 6: **Performance of BC and HBC across baseline environments (average over 8 seeds).** HBC consistently outperforms BC in both success rate and episode length metrics across most environments. In some of the SimpleTown environments, the differences between HBC and BC are negligible, as these tasks are easier to learn and provide limited room for improvement.

Given its efficiency in managing complex environments, HBC was chosen as the backbone for the HILOW framework. By separating high-level and low-level subspaces, HILOW further enhances task adaptation while avoiding unnecessary model expansion, making it well-suited for continual learning in dynamic, complex settings.

## D.2 HIERARCHICAL VS. NON-HIERARCHICAL POLICIES IN GOAL-CONDITIONED CRL

*Table 7* consistently demonstrate that HBC improves over BC, notably in terms of performance (PER) across all CRL baselines tested on both the PointMaze-1 and AntMaze-1 task streams. The most notable improvements are observed in sophisticated methods like FTN, SCN, and PNN, where HBC achieves near-perfect scores, such as 99.4 in PointMaze-1's PNN compared to BC's 96.9.

| Task Stream | CRL Method | PER ↑ | | MEM ↓ | |
|---|---|---|---|---|---|
| | | BC | HBC | BC | HBC |
| PointMaze-1 | EWC | $53.7 \pm 13.7$ | $\mathbf{55.1} \pm 2.9$ | $\mathbf{1.0} \pm 0.0$ | $1.1 \pm 0.0$ |
| | FT1 | $\mathbf{61.4} \pm 16.4$ | $50.0 \pm 2.8$ | $\mathbf{1.0} \pm 0.0$ | $1.1 \pm 0.0$ |
| | FTN | $95.0 \pm 0.9$ | $\mathbf{99.1} \pm 0.8$ | $\mathbf{4.0} \pm 0.0$ | $4.3 \pm 0.0$ |
| | FZ | $\mathbf{41.3} \pm 5.4$ | $34.2 \pm 2.6$ | $\mathbf{1.0} \pm 0.0$ | $1.1 \pm 0.0$ |
| | L2 | $\mathbf{61.3} \pm 6.2$ | $57.4 \pm 6.7$ | $\mathbf{1.0} \pm 0.0$ | $1.1 \pm 0.0$ |
| | PNN | $96.9 \pm 0.1$ | $\mathbf{99.4} \pm 0.8$ | $\mathbf{9.9} \pm 0.0$ | $10.6 \pm 0.0$ |
| | SC1 | $47.0 \pm 5.9$ | $\mathbf{32.3} \pm 5.1$ | $\mathbf{1.0} \pm 0.0$ | $1.1 \pm 0.0$ |
| | SCN | $93.2 \pm 2.8$ | $\mathbf{98.0} \pm 1.1$ | $\mathbf{4.0} \pm 0.0$ | $4.3 \pm 0.0$ |
| AntMaze-1 | EWC | $11.0 \pm 5.9$ | $\mathbf{18.2} \pm 3.1$ | $\mathbf{0.9} \pm 0.0$ | $1.0 \pm 0.0$ |
| | FT1 | $9.2 \pm 2.5$ | $\mathbf{18.3} \pm 1.6$ | $\mathbf{0.9} \pm 0.0$ | $1.0 \pm 0.0$ |
| | FTN | $54.0 \pm 3.1$ | $\mathbf{71.1} \pm 5.1$ | $\mathbf{3.7} \pm 0.0$ | $4.0 \pm 0.0$ |
| | FZ | $19.2 \pm 2.5$ | $\mathbf{24.3} \pm 0.9$ | $\mathbf{0.9} \pm 0.0$ | $1.0 \pm 0.0$ |
| | L2 | $4.6 \pm 2.8$ | $\mathbf{12.3} \pm 3.0$ | $\mathbf{0.9} \pm 0.0$ | $1.0 \pm 0.0$ |
| | PNN | $60.8 \pm 7.4$ | $\mathbf{79.0} \pm 3.9$ | $\mathbf{9.2} \pm 0.0$ | $10.0 \pm 0.0$ |
| | SC1 | $11.3 \pm 2.3$ | $\mathbf{18.0} \pm 1.7$ | $\mathbf{0.9} \pm 0.0$ | $1.0 \pm 0.0$ |
| | SCN | $54.0 \pm 5.0$ | $\mathbf{70.8} \pm 1.9$ | $\mathbf{3.7} \pm 0.0$ | $4.0 \pm 0.0$ |

Table 7: **Performances of BC and HBC on each of the baseline methods (avg. on 3 seeds).** HBC consistently outperforms BC on PER across nearly all CRL methods, with significant gains in more sophisticated approaches such as PNN. Notably, HBC shows superior performance even for challenging methods like EWC and L2, while being only less than 10% more expensive in terms of memory usage. The only exceptions are a few naive and underperforming methods, where the gap is small. This demonstrates HBC as a more effective approach for CRL.

Although HBC introduces a small increase in memory usage (MEM), typically less than $10\%$, this trade-off is minimal compared to the significant performance gains. Even for simpler methods like EWC and L2, HBC demonstrates better PER scores, indicating enhanced retention of previously learned tasks and better adaptation to new ones, which is a key requirement for continual reinforcement learning (CRL).

In both task streams, particularly in more complex settings such as AntMaze-1, HBC manages to reduce catastrophic forgetting and outperform BC consistently. This analysis confirms that HBC offers substantial improvements for CRL across all tested baselines, making it a strong candidate for scaling up to more challenging and dynamic environments.

D.3 HIERARCHICAL GOAL-CONDITIONED CRL BENCHMARK

| Task Stream | CRL Method | PER ↑ | BWT ↑ | FWT ↑ | MEM ↓ |
|---|---|---|---|---|---|
| PointMaze-1 | EWC | 55.1 ± 2.9 | -43.5 ± 3.0 | 0.6 ± 2.3 | 1.0 ± 0.0 |
| | FT1 | 50.0 ± 2.8 | -49.1 ± 3.6 | 1.1 ± 1.9 | 1.0 ± 0.0 |
| | FTN | 99.1 ± 0.8 | 0.0 ± 0.0 | 1.1 ± 1.9 | 4.0 ± 0.0 |
| | FZ | 34.2 ± 2.6 | 0.0 ± 0.0 | -63.8 ± 1.6 | 1.0 ± 0.0 |
| | L2 | 57.4 ± 6.7 | -39.3 ± 6.5 | -1.3 ± 0.2 | 1.0 ± 0.0 |
| | PNN | 99.4 ± 0.8 | 0.0 ± 0.0 | 1.4 ± 1.5 | 9.9 ± 0.0 |
| | SC1 | 32.3 ± 5.1 | -65.7 ± 5.8 | 0.0 ± 0.0 | 1.0 ± 0.0 |
| | SCN | 98.0 ± 1.1 | 0.0 ± 0.0 | 0.0 ± 0.0 | 4.0 ± 0.0 |
| | HILOW (ours) | 98.0 ± 0.4 | 0.0 ± 0.0 | 0.0 ± 0.0 | 2.3 ± 0.0 |
| PointMaze-2 | EWC | 59.1 ± 3.3 | -40.5 ± 3.5 | -0.4 ± 0.7 | 1.0 ± 0.0 |
| | FT1 | 56.1 ± 4.2 | -43.5 ± 4.6 | -0.4 ± 0.7 | 1.0 ± 0.0 |
| | FTN | 99.6 ± 0.7 | 0.0 ± 0.0 | -0.4 ± 0.7 | 4.0 ± 0.0 |
| | FZ | 32.3 ± 2.8 | 0.0 ± 0.0 | -67.7 ± 2.8 | 1.0 ± 0.0 |
| | L2 | 55.2 ± 3.4 | -43.2 ± 4.9 | -1.6 ± 1.5 | 1.0 ± 0.0 |
| | PNN | 99.5 ± 0.9 | 0.0 ± 0.0 | -0.5 ± 0.9 | 9.9 ± 0.0 |
| | SC1 | 55.5 ± 2.5 | -44.5 ± 2.5 | 0.0 ± 0.0 | 1.0 ± 0.0 |
| | SCN | 100.0 ± 0.0 | 0.0 ± 0.0 | 0.0 ± 0.0 | 4.0 ± 0.0 |
| | HILOW (ours) | 99.8 ± 0.4 | 1.6 ± 2.7 | -1.8 ± 2.5 | 1.9 ± 0.1 |

Table 8: **CRL Benchmark for Hierarchical Policies on PointMaze Streams (on 3 seeds).**

| Task Stream | CRL Method | PER ↑ | BWT ↑ | FWT ↑ | MEM ↓ |
|---|---|---|---|---|---|
| AntMaze-1 | EWC | 18.2 ± 3.1 | 0.0 ± 0.0 | -1.9 ± 0.6 | 1.0 ± 0.0 |
| | FT1 | 18.3 ± 1.6 | -52.8 ± 3.6 | -3.4 ± 1.1 | 1.0 ± 0.0 |
| | FTN | 71.1 ± 5.1 | 0.0 ± 0.0 | -3.4 ± 1.9 | 4.0 ± 0.0 |
| | FZ | 24.3 ± 0.9 | 0.0 ± 0.0 | -50.2 ± 1.6 | 1.0 ± 0.0 |
| | L2 | 12.3 ± 3.0 | 0.0 ± 0.0 | -10.8 ± 0.2 | 1.0 ± 0.0 |
| | SC1 | 18.0 ± 1.7 | -56.5 ± 4.0 | 0.0 ± 0.0 | 0.0 ± 0.0 |
| | SCN | 70.8 ± 1.9 | 0.0 ± 0.0 | 0.0 ± 0.0 | 4.0 ± 0.0 |
| | PNN | 79.0 ± 3.9 | 0.0 ± 0.0 | 4.5 ± 1.4 | 10.0 ± 0.0 |
| | HILOW (ours) | 74.1 ± 3.2 | 0.0 ± 0.0 | -0.4 ± 0.0 | 2.8 ± 0.0 |
| AntMaze-2 | EWC | 42.5 ± 5.7 | 0.0 ± 0.0 | 10.3 ± 8.1 | 1.0 ± 0.0 |
| | FT1 | 44.5 ± 6.6 | -41.7 ± 5.8 | 14.4 ± 6.1 | 1.0 ± 0.0 |
| | FTN | 72.8 ± 5.3 | 0.0 ± 0.0 | 1.1 ± 7.6 | 4.0 ± 0.0 |
| | FZ | 24.1 ± 1.6 | 0.0 ± 0.0 | -55.1 ± 12.8 | 1.0 ± 0.0 |
| | L2 | 38.3 ± 6.0 | 0.0 ± 0.0 | 2.8 ± 5.7 | 1.0 ± 0.0 |
| | SC1 | 30.3 ± 2.0 | -41.4 ± 3.2 | 0.0 ± 0.0 | 0.0 ± 0.0 |
| | SCN | 71.7 ± 3.5 | 0.0 ± 0.0 | 0.0 ± 0.0 | 4.0 ± 0.0 |
| | PNN | 85.5 ± 2.4 | 0.0 ± 0.0 | 13.8 ± 2.5 | 10.0 ± 0.0 |
| | HILOW (ours) | 76.5 ± 3.0 | 0.0 ± 0.0 | 4.8 ± 2.8 | 4.0 ± 0.0 |

Table 9: **CRL Benchmark for Hierarchical Policies on AntMaze Streams (on 3 seeds).**

| Task Stream | CRL Method | PER ↑ | BWT ↑ | FWT ↑ | MEM ↓ |
|---|---|---|---|---|---|
| | FT1 | 59.5 ± 9.8 | -28.6 ± 8.6 | 3.6 ± 7.3 | 1.0 ± 0.0 |
| | FTN | 87.7 ± 2.6 | 0.0 ± 0.0 | 4.0 ± 7.1 | 4.0 ± 0.0 |
| | FZ | 54.7 ± 2.7 | 0.0 ± 0.0 | -29.2 ± 9.4 | 1.0 ± 0.0 |
| | PNN | 85.8 ± 2.1 | 0.0 ± 0.0 | 1.4 ± 8.5 | 10.0 ± 0.0 |
| VideoGame-1 | SC1 | 53.6 ± 4.2 | -30.9 ± 5.7 | 0.0 ± 0.0 | 1.0 ± 0.0 |
| | SCN | 82.8 ± 7.2 | 0.0 ± 0.0 | 0.0 ± 0.0 | 4.0 ± 0.0 |
| | EWC | 65.1 ± 4.0 | -22.8 ± 5.5 | 3.4 ± 7.8 | 1.0 ± 0.0 |
| | L2 | 64.6 ± 5.6 | -15.2 ± 6.2 | -4.7 ± 9.5 | 1.0 ± 0.0 |
| | HILOW | 87.8 ± 3.5 | 0.0 ± 0.0 | 3.3 ± 9.2 | 2.6 ± 0.0 |
| | FT1 | 63.7 ± 6.9 | -26.7 ± 9.3 | 6.2 ± 1.1 | 1.0 ± 0.0 |
| | FTN | 90.5 ± 2.5 | 0.0 ± 0.0 | 6.3 ± 1.7 | 1.7 ± 0.0 |
| | FZ | 45.8 ± 6.1 | 0.0 ± 0.0 | -37.0 ± 2.7 | 2.7 ± 0.0 |
| | PNN | 86.7 ± 1.4 | 0.0 ± 0.0 | 2.1 ± 1.0 | 10.0 ± 0.0 |
| VideoGame-2 | SC1 | 64.0 ± 2.6 | -20.3 ± 5.3 | 0.0 ± 0.0 | 1.0 ± 0.0 |
| | SCN | 84.7 ± 4.0 | 0.0 ± 0.0 | 0.0 ± 0.0 | 4.0 ± 0.0 |
| | EWC | 62.2 ± 1.4 | -27.8 ± 3.1 | 5.8 ± 1.9 | 1.9 ± 0.0 |
| | L2 | 66.5 ± 4.3 | -12.5 ± 5.1 | -5.2 ± 2.7 | 2.7 ± 0.0 |
| | HILOW | 90.2 ± 5.4 | 0.0 ± 0.0 | 5.9 ± 3.3 | 3.3 ± 0.0 |

Table 10: **CRL Benchmark for Hierarchical Policies on Video Game Streams (on 3 seeds).**

### D.4 HIERARCHICAL SUBSPACE OF POLICIES ADAPTATION STUDY

To demonstrate the specific contributions of the hierarchical subspace approach, we conducted additional experiments designed to isolate its impact on performance and memory efficiency. By focusing on task streams with logical progressions, we highlight how our method adapts.

We evaluated our method on specific task streams, allowing us to showcase the benefits of the hierarchical subspace approach. We first compare Fine-Tuning (FTN) and Continual Subspace of Policies (CSPO) with HSPO (HILOW w/o LoRA) to isolate the effect of the hierarchical subspaces.

#### D.4.1 DYNAMIC CHANGES STREAM (ANTMAZE)

In this stream within the AntMaze environment, the agent navigates through tasks with the same maze but changes in action dynamics : `umaze-normal`, `umaze-permute_actions`, `umaze-inverse_actions`, and `umaze-permute_actions`. In this stream, the agent must adapt to varying action dynamics without changes in the maze layout.

| CRL Method | PER (Mean ± Std) ↑ | MEM (Relative) ↓ |
|---|---|---|
| FTN | 93.2 ± 6.3 | 4.0 |
| CSPO | **94.5 ± 6.1** | 3.0 |
| HSPO | **94.5 ± 8.0** | **2.93** |

Table 11: **Performance and Memory Usage in the AntMaze Dynamic Changes Stream.**

As shown in Table 11 and Figure 12, HSPO achieves comparable performance to CSPO while demonstrating better memory efficiency. Notably, CSPO can only adapt when it has already encountered both high and low context. Our method effectively leverages the proposed hierarchical subspace approach, to efficiently adapt to logical progressions and changes within the environment.

This demonstrates the core mechanism of our proposed framework, where high-level policies are not redundantly replicated, showing efficient adaptation to new tasks. Nevertheless, the relative memory savings may appear modest due to the intentionally small size of the high-level networks, which were chosen in order to accelerate training (which is relatively long depending on the sequence of tasks).

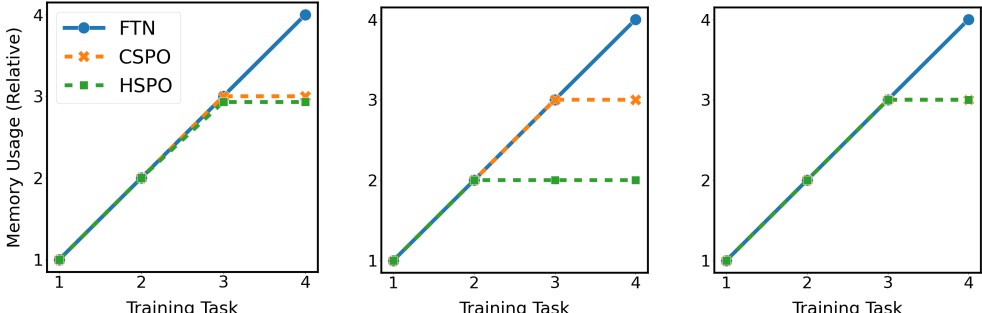

(a) Relative (Total) Memory Size.    (b) Relative (High) Memory Size.    (c) Relative (Low) Memory Size.

Figure 11: Memory Usage in the AntMaze Dynamic Changes Stream.

### D.4.2    TOPOLOGICAL CHANGES STREAM (GODOT MAZE)

This stream examines the agent's ability to adapt to tasks involving changes in maze layouts, requiring the high-level policy to adjust its strategic planning. The tasks included four progressively complex configurations: `maze_1-high`, `maze_2-high`, `maze_3-high`, and `maze_4-high`.

| CRL Method | PER (Mean ± Std) ↑ | MEM (Relative) ↓ |
|:---:|:---:|:---:|
| FTN | **87.2** ± **9.5** | 4.0 |
| CSPO | 87.0 ± 7.8 | 4.0 |
| HSPO | 83.3 ± 12.7 | **3.09** |

Table 12: **Performance and Memory Usage in the Godot Topological Changes Stream.**

As shown in Table 12 and Figure 12, HSPO demonstrates improved memory efficiency over CSPO, with low-level policies able to be selected and reused across tasks. While the performance difference between methods is less pronounced, this stream also highlights the efficient selection process.

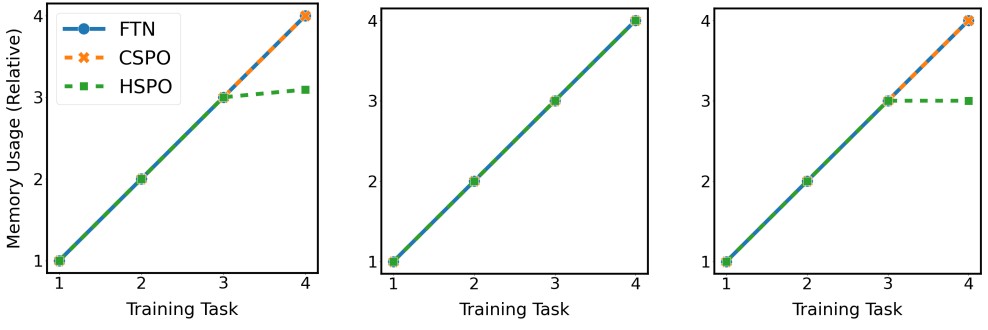

(a) Relative (Total) Memory Size.    (b) Relative (High) Memory Size.    (c) Relative (Low) Memory Size.

Figure 12: Memory Usage in the Godot Topological Changes Stream.

In this case, the relative memory savings are greater due to the larger size of the low-level networks, which are necessary for accurate movements. The hierarchical subspace approach effectively manages policy adaptation and memory usage, particularly in structured task sequences.

### D.4.3 COMPARISON OF CSPO AND HSPO WITH AND WITHOUT LORA

Both CSPO-LoRA and HILOW achieve substantial memory savings due to the use of Low-Rank Adaptation (LoRA). Numerically, LoRA enhances the memory efficiency of our method as Figure 13. The hierarchical subspace approaches (HSPO and HILOW) independently offer performance improvements and additional memory efficiency.

Our hierarchical subspace approach effectively adapts to structured, various changes in the environment, as shown through dynamic and topological task streams.

By separating high-level and low-level policies, HILOW reduces redundancy and improves memory efficiency, aligning with our predictions for specific task streams. Additionally, the integration of Low-Rank Adaptation (LoRA) enhances memory savings, complementing our learning framework.

We believe future work could explore the interpretability of policy adaptations and refine the relationship between LoRA's rank and the nature of environmental changes, which could be interesting for industrial applications, and notably unsupervised hyperparameters tuning.

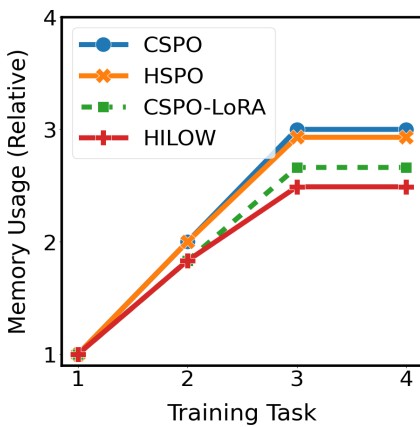

Figure 13: Total Memory Usage in the AntMaze Dynamic Changes Stream.

## E ADDITIONAL EXPERIMENTAL DETAILS

### E.1 ANCHOR WEIGHT SAMPLING

Efficient sampling of anchor weights is essential for exploring a policy subspace. We employ a Dirichlet distributions (Ng et al., 2011) in order to uniformly sample weights within a simplex. Using a symmetric Dirichlet distribution with equal concentration parameters facilitates unbiased exploration across the simplex. To enhance sampling efficiency, we implement the stick-breaking process (Paisley, 2010), which accelerates the generation of anchor weights.

Figure 14 illustrates the effectiveness of our sampling method. Subfigure (a) shows the sampling time in an $N$-dimensional simplex, demonstrating the scalability of our approach. Subfigure (b) displays the coverage of the simplex with three anchors, confirming uniform exploration.

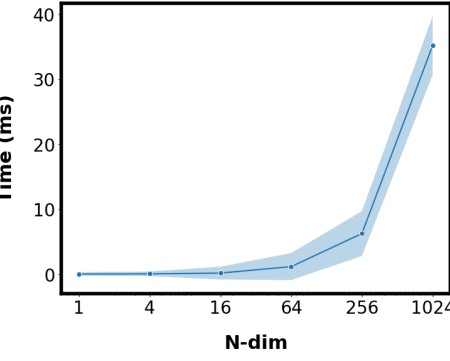
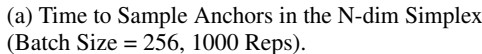
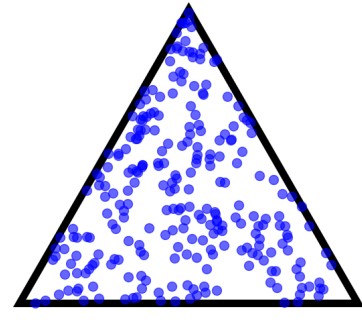

(a) Time to Sample Anchors in the N-dim Simplex (Batch Size = 256, 1000 Reps).

(b) Illustration of the coverage of a 3-Anchor Simplex via Stick Breaking.

Figure 14: Anchor Weight Sampling Illustrations.

While alternative methods, such as gradient-based optimization over the simplex, could be considered, they introduce higher computational costs and risks of converging to local minima. Our

Dirichlet-based sampling method ensures extensive coverage of the weight space with manageable computational overhead, making it well-suited for our offline evaluation framework.

### E.2 COMPUTATIONAL COMPLEXITY AND EFFICIENCY

Evaluating the computational efficiency of our HILOW framework is essential to demonstrate its practicality in continual offline goal-conditioned reinforcement learning. Our experiments across three sets of task streams — PointMaze, AntMaze, and Godot — show that HILOW introduces minimal additional complexity compared to baseline methods, with the main overhead coming from the evaluation of sampled anchor weights within the policy subspace.

In contrast, methods like CSP and PNN incur higher computational costs. CSP slows computation by requiring the learning of a value function, while PNN's ever-growing architecture requires learning connectors to previous layers outputs during both training and inference. These factors result in significant overhead, especially in high-dimensional environments such as Godot.

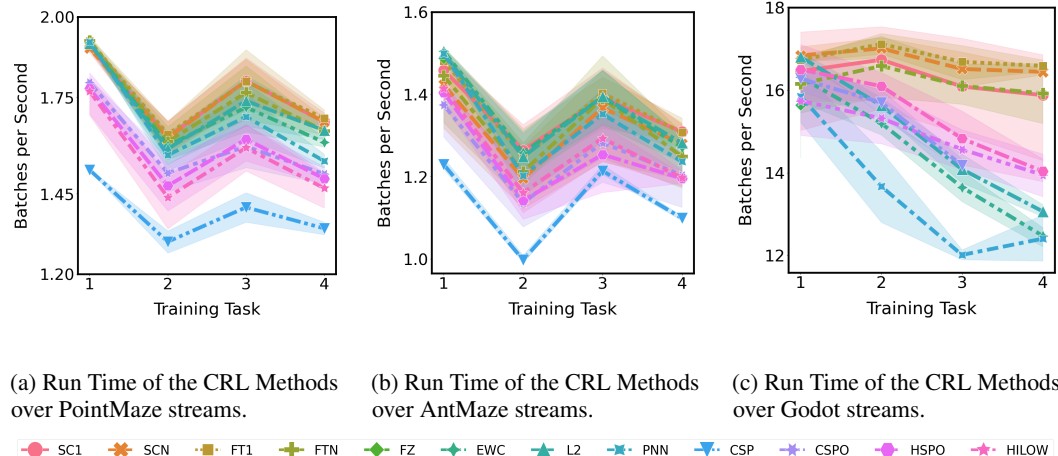

(a) Run Time of the CRL Methods over PointMaze streams.

(b) Run Time of the CRL Methods over AntMaze streams.

(c) Run Time of the CRL Methods over Godot streams.

Figure 15: Anchor Weight Sampling Illustrations.

Figure 15 illustrates the run-time performance of HILOW compared to baseline methods across the task streams, which maintains competitive run-time efficiency.

