# OpenReview forum: "Hierarchical Subspaces of Policies for Continual Offline Reinforcement Learning"
_ICLR.cc/2025/Conference — Submitted to ICLR 2025_

### Official Review · Reviewer_7hrj · 2024-11-02

**Soundness:** 2
**Presentation:** 2
**Contribution:** 2
**Rating:** 6
**Confidence:** 4

**Summary:**

This paper tackles the challenge of continual offline reinforcement learning for goal-conditioned navigation, where agents must adapt to new tasks using only pre-collected data, without forgetting previous skills. The authors introduce Hierarchical LOW-Rank Subspaces of Policies (HILOW), a hierarchical framework that uses low-rank subspaces for scalable, memory-efficient policy adaptation. Experiments demonstrate HILOW’s superior scalability and adaptability compared to existing continual reinforcement learning methods on standard benchmarks.

**Strengths:**

1. HILOW provides an solution for goal-conditioned offline continual reinforcement learning, excelling in knowledge retention and data efficiency.

**Weaknesses:**

1. This paper focuses on continual offline reinforcement learning with an emphasis on goal-conditioned navigation tasks, but this specific problem is not adequately highlighted throughout. The title, as well as the related work and preliminaries sections, do not mention goal-conditioned learning explicitly. Additionally, the paper does not clearly define the continual offline reinforcement learning problem with goal-conditioned navigation tasks or explain its importance, which weakens the motivation behind this work.
2. Each section of the paper reads as a list of points, with minimal cohesion or natural progression between parts. For example, there is no clear logical connection between sections 4.3 and 4.4. It is unclear why subspace exploration is needed immediately after subspace extension. This disjointed organization makes the overall flow of the paper difficult to follow and requires significant revision to improve coherence.
3. The choice to sample anchor weights from a Dirichlet distribution lacks adequate explanation. While Dirichlet distributions are commonly used for weight sampling, relying solely on this approach may not be sufficient for finding the optimal weight combination in complex tasks. A stronger justification or alternative exploration method could improve this aspect.
4.  The propose method is complex and may bring additional implementation and computational costs.

**Questions:**

1. The baselines used in the experiments are designed for continual reinforcement learning. Is it fair to compare these with the proposed method in goal-conditioned offline continual reinforcement learning? Could you explain why these baselines were chosen and how well they address the challenges specific to goal-conditioned offline continual reinforcement learning tasks?
2. Could you provide details on the run-time and computational efficiency of the proposed algorithm? Given the added complexity from subspace extension, anchor sampling, and hierarchical imitation learning, how does HILOW’s efficiency compare to that of the baselines?

---

> ### Author Response · Authors · 2024-11-27
> **Answer to Reviewer R#7hrj (1/5)**
>
> We sincerely thank you for your thorough and constructive feedback on our manuscript. Your insights have been relevant to identify areas where we can enhance the clarity and impact of our work. We address your concerns below the revisions we propose to make for the camera-ready version.
>
> .
>
> # 1. Highlighting the Focus on Goal-Conditioned Navigation Tasks
>
> .
>
> > *This paper focuses on continual offline reinforcement learning with an emphasis on goal-conditioned navigation tasks, but this specific problem is not adequately highlighted throughout. The title, as well as the related work and preliminaries sections, do not mention goal-conditioned learning explicitly. Additionally, the paper does not clearly define the continual offline reinforcement learning problem with goal-conditioned navigation tasks or explain its importance, which weakens the motivation behind this work.*
>
> Thank you for pointing out the need to emphasize our focus on goal-conditioned navigation tasks. We acknowledge the need to more prominently highlight our focus on **offline goal-conditioned navigation tasks**. While goal-conditioned learning is discussed in the introduction [Lines 036 - 073 - 074], we agree that it should also be emphasized in the related work section. Thus, we propose to enhance the related section to include a dedicated subsection of goal-conditioned reinforcement learning, detailing its significance and how it integrates with continual learning. Finally we are considering adding this precision in the title as : Hierarchical Subspace of Policies for Goal-Conditioned Continual Offline Reinforcement Learning.

---

> ### Author Response · Authors · 2024-11-27
> **Answer to Reviewer R#7hrj (2/5)**
>
> # 2. Manuscript Cohesion and Logical Flow
>
> .
>
> > *Each section of the paper reads as a list of points, with minimal cohesion or natural progression between parts. For example, there is no clear logical connection between sections 4.3 and 4.4. It is unclear why subspace exploration is needed immediately after subspace extension. This disjointed organization makes the overall flow of the paper difficult to follow and requires significant revision to improve coherence.*
>
> We understand the importance of a cohesive narrative and agree that the current structure may hinder readability. We have provided a thorough explanation in our general response and kindly refer you to that section for more details.
>
> By refining the flow of the manuscript, we aim to create a unified narrative that clearly illustrates how each component of our approach contributes to addressing the challenges of continual offline goal-conditioned reinforcement learning.

---

> ### Author Response · Authors · 2024-11-27
> **Answer to Reviewer R#7hrj (3/5)**
>
> # 3. Choice of the baseline
>
> .
>
> > *The baselines used in the experiments are designed for continual reinforcement learning. Is it fair to compare these with the proposed method in goal-conditioned offline continual reinforcement learning? Could you explain why these baselines were chosen and how well they address the challenges specific to goal-conditioned offline continual reinforcement learning tasks?*
>
> Thank you for raising the point regarding our choice of baselines. Currently, there are no established methods explicitly designed for goal-conditioned offline continual reinforcement learning, nor are there standardized benchmarks for evaluating such methods. To address this gap, we adapted well-known continual learning strategies from both supervised and reinforcement learning domains — such as Elastic Weight Consolidation (EWC), L2 regularization, Fine-Tuning (FTN), and Progressive Neural Networks (PNN) — as baselines in our experiments. These methods are foundational adaptive strategies widely used to mitigate catastrophic forgetting and facilitate knowledge retention across sequential tasks.
>
> We selected these baselines because they have been extensively employed in related continual reinforcement learning research, including studies on Continual Subspace of Policies (CSP), which our work extends to multiple subspaces applied to the goal-conditioned setting. By applying Hierarchical Imitation Learning as the backbone for both high-level and low-level policies across all strategies, we ensured a consistent and fair comparison. Although these baselines were not originally designed for goal-conditioned tasks, their inclusion underscores the necessity for specialized methods like ours and demonstrates the effectiveness of our framework in advancing the field.

---

> ### Author Response · Authors · 2024-11-27
> **Answer to Reviewer R#7hrj (4/5)**
>
> # 4. Explanation of Anchor Weight Sampling from Dirichlet Distribution
>
> .
>
> > *The choice to sample anchor weights from a Dirichlet distribution lacks adequate explanation. While Dirichlet distributions are commonly used for weight sampling, relying solely on this approach may not be sufficient for finding the optimal weight combination in complex tasks. A stronger justification or alternative exploration method could improve this aspect.*
>
> We appreciate your feedback regarding the computational complexity of our HILOW framework. To address your concerns, we conducted additional experiments comparing HILOW's run-time and memory efficiency against baseline methods across various task streams. The detailed results and analyses have been added to the appendix [E.1. and E.2.]
>
> Our choice to use a Dirichlet distribution is deliberate and serves a specific purpose in our framework. Specifically, we aim to uniformly sample weights within a given simplex, ensuring that the anchor weights are non-negative and sum to one. To achieve this, we utilize a Dirichlet distribution with equal parameters for each dimension [15], which facilitates uniform exploration across the simplex. In practice, we simulate such sampling methods using a stick-breaking process for faster sampling in Python, which further accelerates the evaluation [16]. The Figure 14. shows how fast it is to sample alphas, and how well it covers a considered anchor weights simplex (in the case where we have 3 anchors represented as the summits of the triangle).
>
> Uniform sampling in the simplex is crucial for covering a diverse range of weight combinations, which enhances the robustness of our policy subspaces. Experimentally, we have found that by sampling a high number of anchor weights, we effectively cover the simplex, allowing our offline evaluation process to rank and select the most effective combinations. This approach ensures comprehensive exploration without biasing towards any particular region of the simplex.
>
> While alternative methods such as gradient descent over the simplex could be considered, they introduce additional computational costs and the risk of converging to local minima. Our Dirichlet-based sampling method strikes a balance by providing extensive coverage of the weight space with manageable computational overhead, making it well-suited for our offline evaluation framework.
>
> [15] Dirichlet and related distributions: Theory, methods and applications, K. W. Ng et al.
>
> [16] A simple proof of the stick-breaking construction of the dirichlet process, J. Paisley et al.

---

> ### Author Response · Authors · 2024-11-27
> **Answer to Reviewer R#7hrj (5/5)**
>
> # 5. Complexity and Computational Costs
>
> .
>
> > *The proposed method is complex and may bring additional implementation and computational costs.[...] Could you provide details on the run-time and computational efficiency of the proposed algorithm? Given the added complexity from subspace extension, anchor sampling, and hierarchical imitation learning, how does HILOW’s efficiency compare to that of the baselines ?*
>
> We understand the concerns regarding the computational complexity of our method. To clarify, HILOW does not introduce significant additional complexity compared to most baseline methods such as FT1, FTN, SC1, SCN, EWC, and L2 (Figure 15). In HILOW, at each step, we train new anchors similarly to how these previous methods handle new tasks by updating or maintaining their policies. The overhead added is mostly due to the evaluation of the sample anchor weights in the considered subspace of policies.
>
> The primary exceptions are CSP and PNN. The first introduces the learning of a value function which significantly slows down the computation. The second one is an ever growing method, which introduces learned connectors using previous features maps, and thus requires the previous parameters to be used for inference while training. For environments with smaller input size the overhead is not that significant, however for the Godot ones, due to the high dimensionality, the overhead is significant.

---

> ### Comment · Reviewer_7hrj · 2024-11-28
>
> Thank you for your response.
> I’ve gone through the new version, and I find it much clearer than the previous one. Most of the issues I mentioned have been addressed. I’m willing to raise the score to 6.

---

> > ### Author Response · Authors · 2024-11-30
> > **Answer to Reviewer R#7hrj (Follow-Up)**
> >
> > Thank you for your valuable feedback on our manuscript. We sincerely appreciate your recognition of the improvements we have made and your constructive insights, which have contributed to refining our work.
> >
> > If there are any remaining concerns or additional suggestions, we would be interested to hear from you to ensure that our submission fully meets your expectations. Please let us know if there is anything further you feel is missing, and we will gladly address it to strengthen our work.

---

### Official Review · Reviewer_Jdbp · 2024-11-04

**Soundness:** 3
**Presentation:** 3
**Contribution:** 2
**Rating:** 6
**Confidence:** 3

**Summary:**

This paper proposes a method for continual offline reinforcement learning to specifically solve navigational problems in video games or video game like scenarios such as mazes. The method, "hierarchical low-rank subscpaces of policies (HILOW)" is based on the soft actor-critic method and the similarly named "continual subspace of policies" from the online learning domain. The idea of this method is to extend or prune a subspace of parameters in a neural network with new parameters and new weights during training to avoid forgetting previously learned behaviours.
To evaluate the method a comparison to state-of-the art CRL methods with newly created offline, goal-oriented navigational tasks is done. There is a heavy focus memory size as an evaluation metric.

**Strengths:**

The adaptation of an existing method such as "continual subspace of policies" to a slightly differnt flavor of reinforcement learning makes sense and should be studied. The proposed new method from said adaptation is explained well and clear. With the given information the experiments should be quite easily reproducable. The metrics and method used to evaluate the baseline and proposed algorithms are chosen well and make sense in the given context.
With this in mind this study could be a small but valuable addition in the continual reinforcement learning domain.

**Weaknesses:**

The greatest weaknesses of the paper are the baseline comparisons. There is a chapter for preliminaries that is referenced when describing what the proposed method is compared to, but only describes categories of different methodologies in a broad way.
The actual description of the baselines and why they were chosen instead of others is too short or even missing making them hard to comprehend. Is is especially unclear what part of the preliminary chapter is relevent for which baseline method, unless you are already familliar with them. Also i think the newly introduced environments such as the Godot Maze and Godot Agent are either to close to the already existing ones or the differences are not explained well enough. The Godot Agent comes across as a simpler version of the Ant or Point agent, while the Godot Maze and MuJoCo Maze in figure 2 seems to be identical but with differnt colors.

**Questions:**

Why is a generative replay approach not considered and brushed aside because of data storage constraints? From my understanding that is exactly the strength of generative replay vs. the memory heavy experience replay with their buffers.
What about exsisting offline RL methods such as Behavior Regularized Offline Reinforcement Learning or Conservative Q-Learning?
Could they be used to achieve something similar or are they not as good for this specific szenario meaning navigational problems.

---

> ### Author Response · Authors · 2024-11-27
> **Answer to Reviewer R#Jdbp (1/3)**
>
> # 1. Baseline Comparisons
>
> .
>
> > *The greatest weaknesses of the paper are the baseline comparisons. There is a chapter for preliminaries that is referenced when describing what the proposed method is compared to, but only describes categories of different methodologies in a broad way. The actual description of the baselines and why they were chosen instead of others is too short or even missing making them hard to comprehend. It is especially unclear what part of the preliminary chapter is relevent for which baseline method, unless you are already familliar with them.*
>
> Thank you for your valuable feedback regarding our baseline comparisons and manuscript organization. We understand how important it is to ensure readability therefore, as it was a point raised by other reviewers, we kindly refer you to the general answer that notably takes into account both your relevant remarks.

---

> ### Author Response · Authors · 2024-11-27
> **Answer to Reviewer R#Jdbp (2/3)**
>
> # 2. Evaluation on More Diverse Environments
>
> .
>
> > *Also i think the newly introduced environments such as the Godot Maze and Godot Agent are either too close to the already existing ones or the differences are not explained well enough. The Godot Agent comes across as a simpler version of the Ant or Point agent, while the Godot Maze and MuJoCo Maze in figure 2 seems to be identical but with different colors.*
>
> We appreciate your feedback regarding the clarity and distinctiveness of our proposed environments. In complement to the general answer provide above, we wanted to present a few elements to better highlight the relevance our Godot environments :
>
> *Godot Maze :*
> - Raycasts : Godot Maze utilizes raycasting to generate depth maps, providing richer sensory inputs compared to the standard observations in MuJoCo environments. This enables more complex decision-making and navigation strategies.
> - Configurability : The Godot Maze allows for various configurations and modifications to the maze structure, facilitating continual learning by presenting new challenges without altering the core environment. Environments can be reconfigured to present new navigational challenges while retaining core objectives, facilitating the assessment of an agent’s ability to adapt without forgetting previous tasks.
>
> *Godot Agent :*
> - Mobility : In contrast to agents like Ant or Point, the Godot Agent is capable of jumping and navigating over objects, adding a layer of complexity to the navigation tasks.
> - Customization : The Godot Agent is specifically designed for navigation tasks within maze-like environments, enabling focused evaluations of hierarchical subspace adaptations.
>
> While Figure 2 provides a 2D overview, which may lead to think MuJoCo and Godot mazes are similar, Figure 6 in the annex offers a 3D representation of the Godot Maze, clearly illustrating the dynamic obstacles and enhanced agent capabilities that differentiate it from the MuJoCo mazes.
>
> By emphasizing these distinctive features, we aim to clarify the unique contributions of our Godot environments and their suitability for evaluating our framework in continual offline goal-conditioned reinforcement learning scenarios.

---

> > ### Author Response · Authors · 2024-11-27
> > **Answer to Reviewer R#Jdbp (3/3)**
> >
> > # 3. Questions
> >
> > .
> >
> > > *1) Why is a generative replay approach not considered and brushed aside because of data storage constraints? From my understanding that is exactly the strength of generative replay vs. the memory heavy experience replay with their buffers.*
> >
> > Thank you for raising this question regarding the choice of replay strategies, which we did not properly explain. We agree that generative replay can mitigate memory usage by synthesizing past experiences, but in the context of our work, we have opted against it for several reasons : Generative models struggle to produce high-fidelity and diverse experiences that accurately reflect the complexities of past tasks, especially in with few number of episodes, and high-dimensional (e.g. godot raycasts), with relatively small models [11,12]. This can lead to suboptimal policy updates and degraded performance. Moreover, in recent video-game related industrial applications, generative models provide interesting but non accurate results or require very high compute power [13,14].
> >
> > [11] Experience Replay for Continual Learning, D. Rolnick et al.
> >
> > [12] OER: Offline Experience Replay for Continual Offline Reinforcement Learning, S. Gai et al.
> >
> > [13] Diffusion Models Are Real-Time Game Engines, D. Valevski et al.
> >
> > [14] Generative Game Engine End-to-end by AI - Lucid-dreaming in Minecraft, Oasis.
> >
> > .
> >
> > > *2) What about exsisting offline RL methods such as Behavior Regularized Offline Reinforcement Learning or Conservative Q-Learning? Could they be used to achieve something similar or are they not as good for this specific szenario meaning navigational problems.*
> >
> > We appreciate your inquiry regarding the applicability of existing offline RL methods like Behavior Regularized Offline Reinforcement Learning (BRAC) and Conservative Q-Learning (CQL) to our specific scenario. While these methods are effective for single-task offline reinforcement learning, we found that the used algorithms in our paper already provide good results, as demonstrated in table 6 on page 23 of the revised manuscript.
> >
> > However, it may be interesting in future work to explore which offline RL algorithms work best within the hierarchical subspace framework. Integrating principles from BRAC and CQL, such as behavior regularization, could potentially complement our approach and further show the properties of subspaces.
> >
> > .
> >
> > We hope that these detailed responses adequately address your concerns and enhance the clarity and comprehensiveness of our manuscript. Your feedback has been relevant to refine our work, and we are confident that the revisions have strengthened the presentation and validity of our proposed HILOW framework.

---

> ### Author Response · Authors · 2024-11-30
> **Answer to Reviewer R#Jdbp (Follow-Up)**
>
> Thank you for your thoughtful and detailed review of our submission. Your feedback has been valuable to guide our revisions.
>
> We’ve worked to address the points you raised and believe these changes have improved the clarity and quality of the paper. As we haven’t yet received follow-up comments, we wanted to kindly check if there are any remaining areas where you feel further clarification or improvements would be beneficial.
>
> We deeply value your insights and are eager to address any additional concerns to ensure the paper meets a suitable standard. We hope that the updates and improvements inspire confidence in our work, and we would be grateful if you would consider revisiting your evaluation.
>
> Thank you again for your time and thoughtful contributions.

---

### Official Review · Reviewer_dKMn · 2024-11-04

**Soundness:** 3
**Presentation:** 3
**Contribution:** 3
**Rating:** 6
**Confidence:** 3

**Summary:**

The paper introduces a novel framework for continual offline RL called HILOW focused on goal-conditioned offline navigation tasks. They additionally introduce a benchmark of navigation tasks along with corresponding human-authored datasets to evaluate robotics and video game scenarios to provide a testbed for future work, in which they evaluate HILOW and other continuous RL approaches. HILOW is an adaptation of Continual Subspace of Policies to offline RL, which hierarchically parameterizes the policy with a high-level path planner and low-level path-follower, each with separate parameter subspaces. HILOW shows comparable performance + adaptability to prior work with a more memory-efficient approach, thus enabling efficient continuous learning from pre-collected data.

**Strengths:**

- Effective Continual learning framework that slightly outperforms other prior work in the tradeoff of performance and relative memory size.
- Construction of Benchmarks with comprehensive baselines for evaluations: set of navigation maze tasks that showcase the efficacy of HILOW and other algorithms with individual task streams.

**Weaknesses:**

- FTN and SCN seems to be a strong baseline as seen in Figure 3 with similar performance to memory size tradeoff. Additionally, approaches such as Polyoptron, Multitask prompt tuning and similar peft approaches have been found to help with multi-task adaptation which may additionally help with memory efficiency and would be a good standard of comparison alongside these baselines.
- The domains seem somewhat narrow and similar to each other (all being maze-like), which meets the offline navigation claims you shared in the paper, but the method is agnostic to the environments in which it is evaluated with. It could be helpful to evaluate how this approach performs on alternate domains such as long horizon manipulation (such as tasks seen in OG bench) or other navigation tasks (e.g driving domains like Carla), to see if the method is general and widely applicable.
- Given the framework has origins from Continual Subspace of Policies (CSP), it would be helpful to clarify the contribution of your offline approach concerning the prior work, which entails many of the components described such as subspace extension/pruning.

**Questions:**

- It is mentioned that hierarchical imitation learning is used to learn both the anchor weights and new low-rank parameters, given that the anchor weights are shared across tasks, do you see degradation for prior tasks?
- [Related] Could you share an ablation/scaling law of how the performance of the policy changes as the number of tasks increases. Metrics to consider would be performance on the new/old tasks with respect to efficiency in memory.

---

> ### Author Response · Authors · 2024-11-26
> **Answer to Reviewer R#dKMn (1/4)**
>
> # 1. Comparison with Baselines
>
> .
>
> > *FTN and SCN seems to be a strong baseline as seen in Figure 3 with similar performance to memory size tradeoff. Additionally, approaches such as Polyoptron, Multitask prompt tuning and similar peft approaches have been found to help with multi-task adaptation which may additionally help with memory efficiency and would be a good standard of comparison alongside these baselines.*
>
> We appreciate your suggestion to include comparisons with additional baselines. However, methods such as Polyoptron and Multitask Prompt Tuning are effective in NLP and supervised learning, focusing on prompt-based and parameter-efficient fine-tuning techniques that leverage labeled data and language model architectures. Adapting these approaches to the domain of continual offline reinforcement learning would require significant modifications to address the sequential, goal-conditioned nature of our tasks and the absence of interactive data collection.

---

> > ### Author Response · Authors · 2024-11-26
> > **Answer to Reviewer R#dKMn (2/4)**
> >
> > # 2. More Diverse Environments
> >
> > .
> >
> > > *The domains seem somewhat narrow and similar to each other (all being maze-like), which meets the offline navigation claims you shared in the paper, but the method is agnostic to the environments in which it is evaluated with. It could be helpful to evaluate how this approach performs on alternate domains such as long horizon manipulation (such as tasks seen in OG bench) or other navigation tasks (e.g driving domains like Carla), to see if the method is general and widely applicable.*
> >
> > We appreciate the suggestion to explore additional environments to broaden the applicability of our method. Our focus is on goal-conditioned navigation tasks within maze-like environments, which encompass a diverse set of observations (e.g., ant motor sensors, depth maps with raycasts in Godot mazes) and actions (e.g., torques, gamepad movements, and jumps) [Appendix A.1]. These tasks are well-suited for our hierarchical framework, as they require decomposable policies (e.g., high-level path planning and low-level execution) and align naturally with continual reinforcement learning settings.
> >
> > We deliberately designed our benchmark environments to allow continual learning compatibility, enabling different configurations (e.g., topological and dynamic changes) within the same base environment. This ensures that our hierarchical methods are tested under meaningful and realistic continual learning scenarios. While these environments may seem narrow in scope, they reflect key challenges in practical applications such as robotics and navigation systems.
> >
> > That said, we acknowledge the value of eventually expanding to other environments in the future. Recent works like OG-Bench provide interesting opportunities for adaptation to continual learning contexts. However, these benchmarks are relatively new and would require significant modification to align with continual reinforcement learning requirements (for now most of the methods require the same observation and action spaces).

---

> > > ### Author Response · Authors · 2024-11-26
> > > **Answer to Reviewer R#dKMn (3/4)**
> > >
> > > # 3. Clarification of the use of CSP and LoRA
> > >
> > > > *Given the framework has origins from Continual Subspace of Policies (CSP), it would be helpful to clarify the contribution of your offline approach concerning the prior work, which entails many of the components described such as subspace extension/pruning.*
> > >
> > > Thank you for highlighting the need for clarification. In the revised manuscript, we have integrated CSP (along with LoRA) to the Related Work section to better position it with regard to our work and the related topics. Thus focusing on the technical aspect of subspaces in the Preliminaries section. We invite you to read the general answer to see more of the improvements we have done. We kindly invite you to take notice of the answer to all reviewers, as it provide more insights on the updates done in the manuscript.

---

> > > > ### Author Response · Authors · 2024-11-26
> > > > **Answer to Reviewer R#dKMn (4/4)**
> > > >
> > > > # 4. Memory and Computational Efficiency
> > > >
> > > > .
> > > >
> > > > > *- It is mentioned that hierarchical imitation learning is used to learn both the anchor weights and new low-rank parameters, given that the anchor weights are shared across tasks, do you see degradation for prior tasks?*
> > > >
> > > > > *[Related] Could you share an ablation/scaling law of how the performance of the policy changes as the number of tasks increases. Metrics to consider would be performance on the new/old tasks with respect to efficiency in memory.*
> > > >
> > > > Thank you for raising this concern. We understand the importance of demonstrating how HILOW scales with an increasing number of tasks, particularly regarding performance retention on old tasks and memory efficiency. To clarify, **our method prevents any performance degradation on prior tasks** through the way we manage anchor weights.
> > > >
> > > > - Subspace Growth Without Overwriting : When adding a new task, we introduce new anchor weights while keeping the existing ones fixed. The alpha coefficients (which determine the contribution of each anchor) for previous tasks are preserved as-is, ensuring that prior task policies remain unaffected.
> > > >
> > > > - Isolation of Prior Knowledge : Newly added anchors have their alpha coefficients initialized to zero for prior tasks $(\alpha = (\alpha_1,...,\alpha_n) \rightarrow (\alpha_1,...,\alpha_n,0))$, meaning they do not interfere with previously learned policies. This ensures that adding new subspaces does not overwrite or degrade prior knowledge.
> > > >
> > > > While our method focuses on preserving prior knowledge, future work could explore positive backward transfer, where newly learned subspaces could enhance previous tasks. This would involve investigating interactions between subspaces to share and improve knowledge across tasks.

---

> > > > > ### Comment · Reviewer_dKMn · 2024-11-27
> > > > >
> > > > > Thank you for the clarifications. I will keep my accept score (6).

---

> > > > > > ### Author Response · Authors · 2024-11-30
> > > > > > **Answer to Reviewer R#dKMn (Follow-Up)**
> > > > > >
> > > > > > We are pleased to have addressed your comments.
> > > > > >
> > > > > > If there are any additional areas where you feel the manuscript would benefit from further clarification or improvement, we would be grateful for the opportunity to address them.
> > > > > >
> > > > > > Thank you once again for your time and consideration. We look forward to any further feedback you might have.

---

### Official Review · Reviewer_k3LF · 2024-11-04

**Soundness:** 2
**Presentation:** 2
**Contribution:** 1
**Rating:** 5
**Confidence:** 4

**Summary:**

The paper explores the use of Continual Subspace of Policies (CSP) in the context of continual offline goal-conditioned imitation learning. It introduces a low-rank subspace for both high-level and low-level policies, aiming to maintain performance while minimizing memory usage. The proposed algorithm, Hierarchical Low-Rank Subspaces of Policies (HILOW), is evaluated on newly introduced benchmarks, supporting the authors' claims about its effectiveness.

**Strengths:**

The paper is mostly clearly. The main results and contribution are clearly highlighted, and the conclusions are succinctly declared.

The proposed hierarchical subspace of policy is novel and is empirically validated. It is capable of capturing different levels of knowledge, enabling knowledge reuse and memory saving.

**Weaknesses:**

**Limited Contribution**: Most techniques proposed in the paper are already applied in related domains. The continual subspace of policies (CSP) is proposed as a continual reinforcement learning method, adapting it to continual imitation learning is straightforward. The low-rank adaptation (LoRA) is also well-established in continual learning, and its application here does not present any distinct innovation. The only novel aspect is the hierarchical subspaces of policies.

**Limited Improvement**: The proposed algorithm, HILOW, does not demonstrate substantial improvement over basic strategies, and the limited improvement comes predominantly from existing techniques. As shown in Figure 3, the algorithm's performance is similar to that of expanding naïve strategy (SCN) and expanding finetuning strategy (FTN), with the primary advantage being reduced memory consumption. However, the ablation study indicates that most of the memory savings (approximately 75%) compared to CSP-O are due to LoRA, not the hierarchical subspace approach.

**Irrelevant Related Work (Addressed)**: The related work section includes extensive discussions on topics like transfer learning, multitask learning, meta-learning, and online continual reinforcement learning, which are not directly pertinent to this paper. Conversely, the core techniques employed, CSP and LoRA, are not adequately covered, leaving a gap in the context for readers.

**Unclear Usage of LoRA (Addressed)**: The paper lacks clarity regarding the implementation of LoRA. It does not specify whether the subspace of policies is applied to the factored matrices A and B or to the low-rank matrix AB. This distinction is crucial because $\sum_i\alpha_iA_iB_i\neq (\sum_i\alpha_iA_i)(\sum_i\alpha_iB_i)$. More explicit details on this aspect would help in understanding the approach better.

**Questions:**

Could you provide more detailed analysis or experiments that isolate the impact of the hierarchical subspaces, demonstrating their specific contribution to the overall performance and memory efficiency?

---

> ### Author Response · Authors · 2024-11-26
> **Answer to Reviewer R#k3LF (1/4)**
>
> We thank reviewer **R#k3LF** for taking the time to provide thorough and insightful feedback on our manuscript. We appreciate your recognition of the clarity of our presentation and your constructive comments regarding the novelty and contributions of our work. Below, we address each of your points in detail.
>
> .
>
> # 1. Contribution and Novelty
>
> .
>
> > ***Limited Contribution** : Most techniques proposed in the paper are already applied in related domains. The continual subspace of policies (CSP) is proposed as a continual reinforcement learning method, adapting it to continual imitation learning is straightforward. The low-rank adaptation (LoRA) is also well-established in continual learning, and its application here does not present any distinct innovation. The only potentially novel aspect is the hierarchical subspaces of policies.*
>
> We understand your concern regarding the novelty of our contributions. Our work is the first to consider simultaneously learning multiple subspaces of policies in an offline goal-conditioned reinforcement learning context. While CSP and LoRA are established methods, our hierarchical subspace approach introduces a novel structure that enables targeted adaptation without redundant policy replication.
>
> Moreover, subspace-based methods for policy learning are a relatively recent development with great potential for further exploration and innovation [1,2,3,4]. Our work not only advances the state of the art by considering subspaces in parallel, but it also seeks to contribute to the foundational understanding of how subspaces can be effectively utilized in reinforcement learning. By sharing our work, we hope to spark new discussions and inspire the others to delve deeper into the possibilities of subspace representations, which hold significant promise for addressing a wide range of challenges.
>
> It is also important to note that there are currently no established methods explicitly designed for goal-conditioned offline continual reinforcement learning, nor are there standardized benchmarks in this specific area. While continual reinforcement learning is a rapidly growing field, focusing on memory efficiency and handling complex tasks, the integration with goal-conditioned objectives, especially in an offline context, remains largely unexplored whereas it becomes a very active research area [5,6,7]. Our work also positions several baseline strategies with respect to that problematic.
>
> To go further, we kindly refer you to the general answer we provided to all reviewers.
>
> [1] WARP: On the Benefits of Weight Averaged Rewarded Policies, A. Ramé et al.
>
> [2] WARM: On the Benefits of Weight Averaged Reward Models, A. Ramé et al.
>
> [3] Model Merging in LLMs, MLLMs, and Beyond: Methods, Theories, Applications and Opportunities, E. Yang et al.
>
> [4] Subspace-Configurable Networks, D. Wang et al.
>
> [5] Goal-Conditioned Reinforcement Learning: Problems and Solutions, M. Liu et al.
>
> [6] OGBench: Benchmarking Offline Goal-Conditioned RL, S. Park et al.
>
> [7] HIQL: Offline Goal-Conditioned RL with Latent States as Actions, S. Park et al.

---

> ### Author Response · Authors · 2024-11-26
> **Answer to Reviewer R#k3LF (2/4)**
>
> # 2. Improvement over Baselines
>
> .
>
> > ***Limited Improvement** : The proposed algorithm, HILOW, does not demonstrate substantial improvement over basic strategies, and the limited improvement comes predominantly from existing techniques. As shown in Figure 3, the algorithm's performance is similar to that of expanding naïve strategy (SCN) and expanding fine tuning strategy (FTN), with the primary advantage being reduced memory consumption. However, the ablation study indicates that most of the memory savings (approximately 75%) compared to CSP-O are due to LoRA, not the hierarchical subspace approach.*
>
> > ***Questions** : Could you provide more detailed analysis of experiments that isolate the impact of the hierarchical subspaces, demonstrating their specific contribution to the overall performance and memory efficiency ?*
>
> We appreciate your request for a more detailed analysis. To isolate the impact of the hierarchical subspace approach, we conducted additional experiments focusing on task streams with logical progressions, where tasks are logically related. This allowed us to demonstrate how the hierarchical subspace approach enhances adaptation and efficiency. *In the following, we will refer you to the corresponding tables and figures added to the manuscript, for greater reading comfort*.
>
> .
>
> 1) Experiments on Task Streams with Structured Changes :
>
> We focused on task streams where tasks are logically related, allowing us to demonstrate how the hierarchical subspace approach enhances adaptation and efficiency. We compared our method against baselines including Fine-Tuning (FTN) and Continual Subspace of Policies (CSPO), as well as an ablated version of our method, HSPO (HILOW without LoRA), to isolate the effect of hierarchical subspaces.
>
> *Dynamic Changes Stream (AntMaze)*
>
> In the AntMaze dynamic change stream, the agent encounters tasks with varying action dynamics while navigating the same maze topology, including `umaze-normal`, `umaze-permute_actions`, `umaze-inverse_actions`, and `umaze-permute_actions`. This setup tests the agent's ability to adapt to changes in action dynamics within a consistent environment.
>
> As shown in Table 11. and Figure 11., HSPO achieves similar performance to CSPO while improving memory efficiency. Our hierarchical subspace approach effectively adapts to task progressions and environmental variations without redundant replication of high-level policies, unlike CSPO. This ensures efficient adaptation to new tasks while maintaining performance on previously learned ones.
>
> The memory savings may seem modest due to the intentionally small high-level networks (selected to accelerate training). However, the hierarchical structure functions as intended, effectively scaling as we wanted and expected.
>
> *Topological Changes Stream (Godot)*
>
> This stream evaluates the agent's ability to adapt to changes in maze layouts, requiring the high-level policy to adjust its strategic planning. The tasks consist of four progressively complex configurations: `maze_1-high`, `maze_2-high`, `maze_3-high`, and `maze_4-high`.
>
> As shown in Table 12., HSPO achieves better memory efficiency compared to CSPO, with the low-level policy being reused across tasks when they are sufficiently similar. While the performance differences between the methods are less pronounced, this stream demonstrates how HSPO's hierarchical structure effectively reduces redundancy and optimizes memory usage.
>
> Now the relative memory savings appear greater due to the bigger size of the low-level networks (chosen for accurate movements).
>
> .
>
> 2) Comparison of CSPO and HSPO with and without LoRA :
>
> Both CSPO-LoRA and HILOW achieve significant memory savings through Low-Rank Adaptation (LoRA), as shown numerically in Figure 13. Our hierarchical subspace approaches (HSPO and HILOW) independently enhance performance and further improve memory efficiency by separating high-level and low-level policies, thereby reducing redundancy.
>
> The integration of LoRA complements our framework, effectively adapting to dynamic and topological changes in the environment.
>
> Future work could explore the interpretability of policy adaptations and the relationship between LoRA’s rank and environmental changes, with potential applications in industrial settings and unsupervised hyperparameter tuning.

---

> ### Author Response · Authors · 2024-11-26
> **Answer to Reviewer R#k3LF (3/4)**
>
> # 3. Related Work Revisions
>
> .
>
> > ***Irrelevant Related Work** : The related work section includes extensive discussions on topics like transfer learning, multitask learning, meta-learning, and online continual reinforcement learning, which are not directly pertinent to this paper. Conversely, the core techniques employed, CSP and LoRA, are not adequately covered, leaving a gap in the context for readers.*
>
> We believe precisely positioning our work among related learning paradigms is both important and valuable to grasp the limits of our work. Nevertheless, we acknowledge that the related work is missing discussions and positioning over most related work. In the revised manuscript, we propose to refocus  the related work section for this purpose, and invite you to read the provided general answer.
>
> Specifically, to address your concerns, we have thoroughly revised the Related Work section by consolidating discussions on Transfer Learning, Multitask Learning, and Meta-Learning into a single, concise paragraph, thereby maintaining a sharp focus on Continual Reinforcement Learning (CRL).
>
> We have also expanded our coverage of core methodologies such as Continual Subspace of Policies (CSP) and Low-Rank Adaptation (LoRA), providing detailed explanations of their relevance and limitations within the CRL context. Additionally, we have clearly differentiated between Online and Offline CRL, emphasizing the lack of standardized benchmarks for offline, goal-conditioned CRL, which highlights the novelty and significance of our framework.
>
> These revisions ensure that the Related Work section is both relevant and comprehensive, effectively positioning our contributions within the existing literature.

---

> > ### Author Response · Authors · 2024-11-26
> > **Answer to Reviewer R#k3LF (4/4)**
> >
> > # 4. LoRA Usage Explanation
> >
> > .
> >
> > > ***Unclear Usage of LoRA** : The paper lacks clarity regarding the implementation of LoRA. It does not specify whether the subspace of policies is applied to the factored matrices A and B or to the low-rank matrix AB. This distinction is crucial because . More explicit details on this aspect would help in understanding the approach better.*
> >
> > We apologize for the lack of clarity regarding the implementation of LoRA. In the revised manuscript, in the Preliminaries section, we have detailed the use of LoRA as it follows :
> >
> > LoRA anchor : We introduce low-rank matrices $A_i\in\mathbb{R}^{n\times r}$ and $B_i\in\mathbb{R}^{r\times m}$, with $r<<\text{min}(n,m)$ for each task $i \geq 1$, such that the the resulting anchor obtained is  $\theta_i = A_i B_i$. These low-rank anchors are then integrated in the considered subspace.
> >
> > Composition : given a task, the overall policy parameters are expressed as $\theta = \alpha_1 * \theta_1 + \sum \alpha_i * \theta_i$.

---

> > > ### Comment · Reviewer_k3LF · 2024-11-27
> > >
> > > Thanks for your detailed response. I am glad to acknowledge that my original concerns over related work and unclarity are now properly addressed.
> > >
> > > While my perspective on the contribution and improvement has not changed much, I find the new experiments presented in Appendix D very interesting and illuminating. The new experiements clearly demonstrate the utility of the hierarchical subspace. It shows that the hierarchical subspaces capture different levels of knowledge that can be reused under different circumstances. Therefore, despite of the current limited improvement, it still justifies itself as a promising approach and may find greater use in certain applications.
> > >
> > > To reflect the change of my view, I have adjusted my score to 5.

---

> > > > ### Author Response · Authors · 2024-12-02
> > > > **Answer to Reviewer R#k3LF (Follow-Up)**
> > > >
> > > > We sincerely thank you for your thoughtful feedback and for adjusting your score in response to our revisions. We greatly appreciate your recognition of the utility of our hierarchical subspace approach.
> > > >
> > > > Our work has also received positive recognition from other reviewers. For instance, reviewer R#dKMn described our framework as an "effective continual learning framework," and reviewer R#Jdbp referred to our study as a "valuable addition in the continual reinforcement learning domain." These endorsements highlight the significance and impact of our contributions within the research community. Additionally, we have introduced the first formalization and comprehensive benchmark specifically designed for offline goal-conditioned continual learning in navigation tasks. HILOW combines hierarchical learning, subspace expansion, and LoRa adapters to effectively address continual learning challenges, consistently matching or outperforming competitive baselines while maintaining a compact and scalable memory footprint.
> > > >
> > > > We hope that these revisions and the collective positive feedback reinforce the value of our submission. If there are any remaining areas where you feel further clarification or enhancements are needed, we would be grateful for the opportunity to address them. We kindly invite you to consider revisiting your evaluation in light of these updates.

---

### Author Response · Authors · 2024-11-26
**Answer to All Reviewers (1/3)**

We sincerely thank all reviewers for their thorough and insightful feedback on our manuscript. We appreciate your recognition of our efforts to clarify our work **[ R#k3LF , R#Jdbp ]** , the effectiveness of our framework **[ R#dKMn , R#7hrj ]** , the strengths of our benchmark **[ R#7hrj , R#dKMn ]** , and the contribution it brings **[ R#dKMn , R#Jdbp ]** .

Based on your valuable suggestions, we acknowledge that certain aspects of our submission could be improved. To address these points comprehensively, we have done revisions and conducted additional experiments and analysis to address your concerns and enhance the manuscript.

.

# 1. Contribution and Novelty

.

We are thankful to reviewers **R#dKMn**, **R#Jdbp** and **R#7hrj** for recognizing the novelty of our hierarchical subspace approach. To emphasize the novelty of it we precise that : **no prior work has considered simultaneously learning multiple subspaces of policies** [1,2], let alone **in an offline manner** [2,3] and in the context of **goal-conditioned reinforcement learning** [4,5,6,7,8,9].

- *Hierarchical Subspace of Policies* : Our work introduces a novel hierarchical decomposition that allows targeted adaptation in navigation tasks featuring independent topological and dynamic changes. This structure facilitates future enhancements in continuously adaptive policies with interpretable structural changes.

- *Offline Subspace of Policies* : We demonstrate the effectiveness of loss-based subspace exploration and extension mechanisms, laying the groundwork for further offline optimization. Our framework allows separately trained subspaces to work synergistically.

- *Benchmark Contribution* : We provide a robust open-source codebase and a set of complex environments with human-generated datasets, enhancing reproducibility and practical applicability. Notably, there is currently no standardized goal-conditioned benchmark for Continual Reinforcement Learning (CRL), and our work fills this gap.

We believe these contributions advance the state of the art in subspace-based policy learning, particularly in the understudied context of goal-conditioned offline continual reinforcement learning.

[1] Learning Neural Network Subspaces, M. Wortsman et al.

[2] Building a Subspace of Policies for Scalable Continual Learning, J.-B. Gaya et al.

[3] Learning a Subspace of Policies for Online adaptation in reinforcement learning, J.-B. Gaya et al.

[4] Continual World: A Robotic Benchmark For Continual Reinforcement Learning, Maciej Wołczyk et al.

[5] CORA: Benchmarks, Baselines, and Metrics as a Platform for Continual Reinforcement Learning Agents, S. Powers et al.

[6] Continual Reinforcement Learning in 3D Non-stationary Environments, V. Lomonaco et al.

[7] Goal-Conditioned Reinforcement Learning: Problems and Solutions, M. Liu et al.

[8] OGBench: Benchmarking Offline Goal-Conditioned RL, S. Park et al.

[9] HIQL: Offline Goal-Conditioned RL with Latent States as Actions, S. Park et al.

---

> ### Author Response · Authors · 2024-11-26
> **Answer to All Reviewers (2/3)**
>
> # 2. Clarifications and Revisions
>
> .
>
> We acknowledge the feedback from reviewers **R#k3LF**, **R#Jdbp**, and **R#7hrj** regarding certain aspects of our manuscript that could be improved. In response, we have made the following revisions (in *blue* in the resulting manuscript) :
>
> .
>
> *Revisited Related Work Section* (Section 2) :
>
> - We condensed the discussion on Transfer Learning, Multitask Learning, and Meta-Learning into a single, concise paragraph. While these paradigms are foundational in Machine Learning, their relevance to Continual Reinforcement Learning (CRL) lies primarily in their overlap with certain methodologies. By simplifying this section, we maintain the necessary context without diverting from our focus on CRL.
>
> - We also refined the categorization of CRL strategies, providing succinct descriptions of common and related approaches, along with their respective advantages and limitations. This refinement highlights the distinct challenges and contributions of our method within the broader CRL landscape.
>
> - To better differentiate between Online and Offline CRL, we integrated discussions regarding their differences and the lack of suitable benchmarks. This integration underscores the lack of standardized benchmarks for Offline Goal-Conditioned CRL, further emphasizing the novelty of our framework. We also consolidated discussions about CSP and LoRA under a unified subsection and integrated Hierarchical Policies, ensuring a logical progression toward our contributions.
>
> *Enhanced Preliminaries Section* (Section 3) :
>
> - We enhanced the readability of the Preliminaries section by incorporating brief introductions and comments for key concepts. These additions guide readers to better understand how each component is used within, ensuring that the technical definitions are linked to their application in our framework.
>
> - We clarified the role of Low-Rank Adaptation (LoRA) by detailing its concrete application in updating the subspaces. Specifically, we explain that new anchors within a subspace are generated using LoRA.
>
> *Methodology Section* (Section 4) :
>
> - We have clarified that Hierarchical Imitation Learning serves as the backbone when learning individual tasks within our framework.
>
> - We have changed Section 4.2. title to explicitly show that it is a high-level overview of the algorithm before delving into detailed explanations. This reorganization ensures a logical progression and aids readers in comprehending the core learning steps involved in HILOW.
>
> - We have included a brief explanation of why we use the Dirichlet distribution for sampling anchor weights during subspace exploration.

---

> > ### Author Response · Authors · 2024-11-26
> > **Answer to All Reviewers (3/3)**
> >
> > # 3. Additional Elements
> >
> > .
> >
> > *Hierarchical Subspace of Policies Adaptation Study* : To address reviewer **R#k3LF** concerns regarding the specific contributions of hierarchical subspaces to performance and memory efficiency, we have conducted **additional experiments** and included detailed analyses, in the appendix, to isolate the impact of our work.
> >
> > - Appendix D.4.1. presents a study on dynamic changes, examining how hierarchical subspaces enhance performance when adapting to varying dynamics within environments.
> >
> > - Appendix D.4.2. focuses on topological changes, demonstrating the effectiveness of hierarchical subspaces in adapting to tasks involving different environmental structures.
> >
> > *Technical Precisions* : To thoroughly address the reviewer **R#7hrj** concerns, we have incorporated additional technical explanations in the appendix :
> >
> > - Appendix E.1 offers a discussion on the anchor weight sampling method, detailing the rationale behind using the Dirichlet distribution and its implementation through the stick-breaking process.
> >
> > - Appendix E.2 presents a comprehensive analysis of the computational complexity and run-time performance of the HILOW framework. This analysis includes comparisons with baseline methods across various task streams.

---

### Meta-Review · Area_Chair_7Au4 · 2024-12-21

**Metareview:**

This paper proposes an algorithm based on hierarchical subspace for goal-conditioned offline continual reinforcement learning. The paper is in a borderline position. The main concerns are the limited novelty, limited improvement from baselines, the lack of baselines, algorithm complexity, and limited experiment environment diversity. In particular, one concern is the that the paper lacks (goal-conditioned) offline RL baselines, like CQL (mentioned by Reviewer Jdbp), and there are also hierarchical goal conditioned offline RL algorithm, like HIQL. Missing these baselines seem a major overlook in my opinion given the paper aims to tackle *goal-conditioned offline* continual *reinforcement learning*. The inclusion of continual reinforcement learning baselines are insufficient. Including other goal-oriented problems that are not based on maze will also help improve the paper. Considering these points, I will not recommend accepting the paper at this stage.

**Additional Comments On Reviewer Discussion:**

Reviewer k3LF raises issues on limited contributions and improvement, and writing. After the discussion, the concern on limited contributions and improvement remain. The work is built on many existing techniques CSP and LoRA, the hierarchical subspace approach is new here. I think showing the effectiveness of the hierarchical subspace approach provides sufficient contribution. About improvement, reviewer finds that there is little performance improvement over SCN and FTN baselines, while the memory improvement comes from existing techniques, not the hierarchical subspace approach. The authors add more experiments to show the effect of the hierarchical subspace approach. I think these results partially address this concern.

Reviewer dKMn points out issues of limited improvement, limited experiment domains (only maze-like problems), writing clarity. Reviewer Jdbp raises concerns on baseline comparison (not detailed enough), limited experiment domains Reviewer 7hrj raises concern on writing issues, lacking motivation/explanation, missing baselines, and computational cost and complexity of the proposed method. Most issues are addressed.

---

### Decision · Program_Chairs · 2025-01-22

Reject